# The JapanFlux2024 dataset for eddy covariance observations covering Japan and East Asia from 1990 to 2023

Masahito Ueyama[1], Yuta Takao[1], Hiromi Yazawa[2], Makiko Tanaka[2], Hironori Yabuki[3], Tomo'omi Kumagai[4], Hiroki Iwata[5], Md. Abdul Awal[6], Mingyuan Du[7], Yoshinobu Harazono[8], Yoshiaki Hata[4], Takashi Hirano[9], Tsutom Hiura[4], Reiko Ide[10], Sachinobu Ishida[11], Mamoru Ishikawa[12], Kenzo Kitamura[13], Yuji Kominami[14], Shujiro Komiya[15], Ayumi Kotani[16], Yuta Inoue[14], Takashi Machimura[17], Kazuho Matsumoto[18], Yojiro Matsuura[14], Yasuko Mizoguchi[19], Shohei Murayama[20], Hirohiko Nagano[21], Taro Nakai[22], Tatsuro Nakaji[23], Ko Nakaya[24], Shinjiro Ohkubo[25], Takeshi Ohta[26,☆], Keisuke Ono[27], Taku M. Saitoh[28], Ayaka Sakabe[29], Takanori Shimizu[14], Seiji Shimoda[30], Michiaki Sugita[31], Kentaro Takagi[32], Yoshiyuki Takahashi[10], Naoya Takamura[4], Satoru Takanashi[19], Takahiro Takimoto[27], Yukio Yasuda[14], Qinxue Wang[10], Jun Asanuma[33], Hideo Hasegawa[21], Tetsuya Hiyama[34], Yoshihiro Iijima[35], Shigeyuki Ishidoya[20], Masayuki Itoh[36], Tomomichi Kato[9], Hiroaki Kondo[20], Yoshiko Kosugi[29], Tomonori Kume[37], Takahisa Maeda[20], Shoji Matsuura[27], Trofim Maximov[38], Takafumi Miyama[14], Ryo Moriwaki[39], Hiroyuki Muraoka[4], Roman Petrov[38], Jun Suzuki[40], Shingo Taniguchi[41], and Kazuhito Ichii[2]

[1]Graduate School of Agriculture, Osaka Metropolitan University, Sakai 599-8531, Japan
[2]Center for Environmental Remote Sensing (CEReS), Chiba University, Chiba 263-8522, Japan
[3]National Institute of Polar Research (NIPR), Tokyo 190-8518, Japan
[4]Graduate School of Agricultural and Life Sciences, The University of Tokyo, Tokyo 113-8657, Japan
[5]Department of Environmental Science, Faculty of Science, Shinshu University, Matsumoto 390-8621, Japan
[6]Department of Crop Botany, Bangladesh Agricultural University, Mymensingh 2202, Bangladesh
[7]Xinjiang Institute of Ecology and Geography, Chinese Academy of Sciences, Xinjiang 830011, China
[8]International Arctic Research Center, University of Alaska Fairbanks, Fairbanks, AK 99775, USA
[9]Research Faculty of Agriculture, Hokkaido University, Sapporo 060-8589, Japan
[10]National Institute for Environmental Studies, Tsukuba 305-8506, Japan
[11]Graduate School of Science and Technology, Hirosaki University, Hirosaki 036-8561, Japan
[12]Faculty of Earth Environmental Science, Hokkaido University, Sapporo 060-0810, Japan
[13]Kyushu Research Center, Forestry and Forest Products Research Institute, Kumamoto 860-0862, Japan
[14]Forestry and Forest Products Research Institute, Tsukuba 305-8687, Japan
[15]Department of Biogeochemical Processes, Max Planck Institute for Biogeochemistry, 07745 Jena, Germany
[16]Graduate School of Bioagricultural Sciences, Nagoya University, Nagoya 464-8601, Japan
[17]Graduate School of Engineering, Osaka University, Suita 565-0871, Japan
[18]Faculty of Agriculture, Iwate University, Morioka 020-8550, Japan
[19]Kansai Research Center, Forestry and Forest Products Research Institute, Kyoto 612-0855, Japan
[20]National Institute of Advanced Industrial Science and Technology (AIST), Tsukuba 305-8569, Japan
[21]Institute of Science and Technology, Niigata University, Niigata 950-2181, Japan
[22]School of Forestry and Resource Conservation, National Taiwan University, Taipei 106319, Taiwan
[23]Sapporo Experimental Forest, Hokkaido University, Sapporo 060-0809, Japan
[24]Sustainable System Research Laboratory, Central Research Institute of Electric Power Industry, Abiko 270-1194, Japan
[25]Forestry Research Institute, Forest Research Department, Hokkaido Research Organization, Bibai 079-0198, Japan
[26]Nagoya University, Nagoya 464-8601, Japan

[27]Institute for Agro-Environmental Sciences, National Agriculture and Food
Research Organization (NARO), Tsukuba 305-8604, Japan
[28]Center for Environmental and Societal Sustainability, Gifu University, Gifu 501-1193, Japan
[29]Graduate School of Agriculture, Kyoto University, Kyoto 606-8502 Japan
[30]Memuro Research Station, Hokkaido Agricultural Research Center, NARO,
(HARC/M /NARO), Memuro 082-0081, Japan
[31]Institute of Life and Environmental Sciences, University of Tsukuba, Tsukuba 305-8572, Japan
[32]Field Science Center for Northern Biosphere, Hokkaido University, Toikanbetsu, 098-2943, Japan
[33]Center for Research in Isotopes and Environmental Dynamics (CRiED),
University of Tsukuba, Tsukuba 305-8572, Japan
[34]Institute for Space-Earth Environmental Research, Nagoya University, Nagoya 464-8601, Japan
[35]Department of Geography, Tokyo Metropolitan University, Tokyo 192-0397 Japan
[36]Research Institute for Sustainable Humanosphere, Kyoto University, Uji 611-0011, Japan
[37]Kasuya Research Forest, Kyushu University, Fukuoka 811-2415, Japan
[38]Institute for Biological Problems of Cryolithozone, Yakutsk 677980, Russia
[39]Faculty of Engineering, Ehime University, Matsuyama 790-8577, Japan
[40]Faculty of Agriculture, Shinshu University, Nagano 399-4598, Japan
[41]Faculty of Agriculture, University of the Ryukyus, Okinawa 903-0213, Japan
☆retired

**Correspondence:** Masahito Ueyama (mueyama@omu.ac.jp)

Received: 28 December 2024 – Discussion started: 12 February 2025
Revised: 20 May 2025 – Accepted: 20 May 2025 – Published:

**Abstract.** Eddy covariance observations play a pivotal role in understanding the land–atmosphere exchange of energy, water, carbon dioxide ($CO_2$), and other trace gases, as well as the global carbon cycle and earth system. To promote the networking of individual measurements and the sharing of data, FLUXNET links regional networks of researchers studying land–atmosphere processes. JapanFlux was established in 2006 as a national branch of AsiaFlux. Despite the growing amount of shared data globally, the availability in Asia is currently limited. In this study, we developed an open dataset of the eddy covariance observations for Japan and East Asia, called JapanFlux2024, that was conducted by researchers affiliated with Japanese research institutions. The data were processed using selected standard methods from the FLUXNET community, with adaptations specific to the JapanFlux2024 dataset. Here, we present the data description and data processing and show the value of processed fluxes of sensible heat, latent heat, and $CO_2$. The dataset will facilitate important studies for Japan and East Asia, such as land–atmosphere interactions, improvement of process models, and upscaling fluxes using machine learning and remote sensing technology, as well as bridge collaborations between Asia and FLUXNET. TS1

# 1 Introduction

The global network of micrometeorological flux observations, FLUXNET (Delwiche et al., 2024; https://fluxnet.org/, last access: 27 December 2024), plays a pivotal role in multidisciplinary fields, such as land–atmosphere interactions, global biogeochemical cycles, and earth system science (Baldocchi et al., 2024; Bonan et al., 2012). FLUXNET started in 1997 as a global network of eddy covariance observations that provides data on land–atmosphere exchanges of energy, water, carbon dioxide ($CO_2$), methane ($CH_4$), and other trace gases by measuring direct turbulent transfer. The quasi-continuous eddy covariance observations revealed variations of land–atmosphere exchange at the diurnal, seasonal, inter-annual, and decadal scales, ranging from site (Takamura et al., 2023; Ueyama et al., 2024) to global (Beer et al., 2010; Keenan et al., 2023; Ueyama et al., 2020a) scales.

The eddy flux communities have developed publicly open databases to promote the multidisciplinary sciences. FLUXNET has periodically released the open datasets for eddy covariance observations: La Thuile Database (252 sites in 2007; Verma et al., 2014; https://fluxnet.org/data/la-thuile-dataset/, last access: 27 December 2024) and FLUXNET2015 (212 sites in 2015; Pastorello et al., 2020). Together with the global carbon project (Friedlingstein et al., 2023; https://www.globalcarbonproject.org, last access: 27 December 2024), FLUXNET also provided a topical dataset, FLUXNET-$CH_4$ (Delwiche et al., 2021), which pro-

motes understanding of wetland $CH_4$ emissions across the globe (Knox et al., 2019; Ueyama et al., 2023). Multiple open databases for the environmental sciences have also been developed for understanding $CO_2$ fluxes in high-latitude ecosystems (Virkkala et al., 2022) and soil respiration (Bond-Lamberty et al., 2020).

Asia has ca. ∼ 60 % of the total world population, and thus humans have been intensively modifying forest land cover in this region for food and energy production. Such land use changes, in combination with climate change, are likely to impact the regional and global carbon and water cycling. These issues are the greatest environmental concerns for the survival of the human population. Flux studies using eddy covariance observations were conducted since the early 1990s for agricultural fields, wetlands, lakes, plantations, primary and secondary forests, disturbed ecosystems, and urban areas. In Asia, although private databases for eddy covariance measurements were developed (Hirata et al., 2008; Ichii et al., 2017; Saigusa et al., 2013), no open databases have yet been developed except the AsiaFlux database (https://asiaflux.net/, last access: 27 December 2024), which does not provide consistent gap-filling and flux partitioning.

JapanFlux (https://www.japanflux.org/, last access: 27 December 2024) was established in 2006 as a national branch of AsiaFlux (Kang and Cho, 2021; Mizoguchi et al., 2009) for the promotion of a network of micrometeorological measurements by researchers affiliated with Japanese research institutions. The mission of JapanFlux is to promote micrometeorological measurements and their collaborations with each other, researchers from other countries, and other research fields (e.g., remote sensing and modeling). Measurements by Japanese institutions have been conducted in Japan and other regions of East Asia (Mizoguchi et al., 2009; Saigusa et al., 2013) since the early 1990s for understanding energy, water, carbon, and greenhouse gas exchanges at various land surfaces.

In this study, we developed JapanFlux2024, the first publicly open dataset by JapanFlux that consists of micrometeorological data measured since the early 1990s. The data are processed with the selected standard methods employed by the FLUXNET community. The dataset is prepared with consistent post-processing, such as gap-filling and flux partitioning, and provides data at various temporal resolutions of half-hourly/hourly, daily, weekly, monthly, and annual intervals. The dataset consists of data collected at 83 sites with 683 site-years. The dataset promotes collaborations between researchers in Japan and other countries and improves our understanding of land–atmosphere interactions.

## 2 Data and methods

The JapanFlux2024 dataset is processed using selected standard methods from the FLUXNET community, with adaptations specific to the JapanFlux2024 dataset. According to

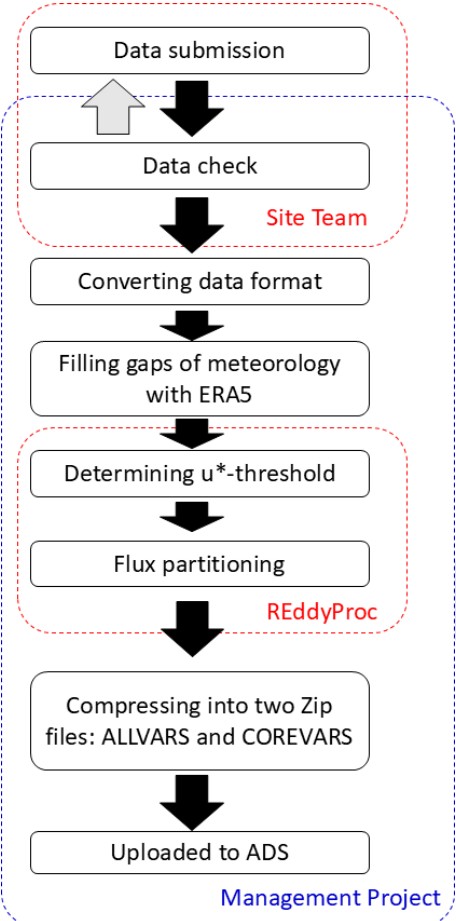

**Figure 1.** Flow chart of data processing in the JapanFlux2024 dataset. The details in each step and meaning of abbreviations are shown in the text.

the processing strategy of Pastorello et al. (2020), the Japan-Flux2024 dataset was developed in four steps: (1) data submission by site teams, (2) formatting data in a FLUXNET format, (3) gap-filling and flux partitioning, and (4) preparing subsets and complete datasets (Fig. 1). Meta data files, so-called Biological, Ancillary, Disturbance, and Metadata (BADM), were also prepared. The data are available from the data portal (https://ads.nipr.ac.jp/japan-flux2024/, last access: 27 December 2024) under the Arctic and Antarctic Data archive System (ADS). Under the ADS, a digital object identifier (DOI) was provided for each site (Table 1). The processing pipeline mentioned in this data paper represents steps downstream of "Filling gaps in meteorology with ERA5" in Fig. 1.

### 2.1 Data collections

We collected the micrometeorological measurement data from the site teams, which were identified using the web pages for AsiaFlux (https://www.asiaflux.net/, last access: 11

**Table 1.** [TS2] Information about sites included in the JapanFlux2024 dataset.

| Site code (BADM) | AsiaFlux ID | Country | Country ID | Site name | Latitude (degree) | Longitude (degree) | Elevation (m) | Köppen climate | IGBP (land use) | Status | Years | Reference | Data citation |
|---|---|---|---|---|---|---|---|---|---|---|---|---|---|
| RU-Tur | TUR | Russia | RU | Tura | 64.208888 | 100.463555 | 250 | Dfc | DNF | Ongoing | 2004 | Nakai et al. (2008) | Matsuura and Morishita (2025) |
| RU-NeB | | Russia | RU | Neleger Burnt Forest | 62.325937 | 129.487342 | 221 | Dfd | GRA | Completed | 1999–2000 | Iwahana et al. (2005) | Machimura (2025a) |
| RU-NeF | | Russia | RU | Neleger larch forest | 62.315615 | 129.499964 | 223 | Dfd | DNF | Completed | 1999–2006 | Iwahana et al. (2005) | Machimura (2025b) |
| RU-NeC | | Russia | RU | Neleger Cutover | 62.314844 | 129.500075 | 221 | Dfd | OSH | Completed | 2001–2006 | Iwahana et al. (2005) | Machimura (2025c) |
| RU-SkP | YLF | Russia | RU | Yakutsk Spasskaya Pad larch | 62.25471 | 129.618543 | 217 | Dfc | DNF | Ongoing | 2004–2014 | Ohta et al. (2008) | Maximov et al. (2025b) |
| RU-Sk2 | YPF | Russia | RU | Yakutsk Spasskaya Pad Pine | 62.241291 | 129.651336 | 216 | Dfc | ENF | Completed | 2004–2008 | Hamada et al. (2004) | Kotani et al. (2025) |
| RU-USk | | Russia | RU | Ulakhan Sykkhan Alas | 62.150995 | 130.527517 | 143 | Dfd | GRA | Completed | 2000 | Yabuki et al. (2004) | Yabuki et al. (2025) |
| RU-Ege | | Russia | RU | Elgeeii forest station | 60.01551563 | 133.8240123 | 203 | Dfd | DNF | Ongoing | 2010–2018 | Kotani et al. (2014) | Maximov et al. (2025a) |
| MN-Skt | SKT | Mongolia | MN | Southern Khentei Taiga | 48.351861 | 108.654333 | 1630 | Dwc | DNF | Completed | 2003–2006 | Li et al. (2005b) | Asanuma (2025b) |
| MN-Udg | | Mongolia | MN | Udleg practice forest | 48.25638888 | 106.8511111 | 1342 | Dwc | DNF | Ongoing | 2010–2012 | Miyazaki et al. (2014) | Ishikawa (2025) |
| MN-Nkh | | Mongolia | MN | Nalaikh grassland | 47.693592 | 107.489342 | 1531 | BSk | GRA | Completed | 2015–2020 | Wang et al. (2023) | Wang et al. (2025b) |
| MN-Hst | | Mongolia | MN | Hustai grassland | 47.594131 | 105.856439 | 1227 | BSk | GRA | Completed | 2015–2020 | Wang et al. (2023) | Wang et al. (2025a) |
| MN-Kbu | KBU | Mongolia | MN | Kherlenbayan Ulaan | 47.213972 | 108.737333 | 1235 | Bsk | GRA | Completed | 2003–2009 | Li et al. (2005a) | Asanuma (2025a) |
| CN-Lsh | LSH | China | CN | Laoshan | 45.279839 | 127.578206 | 340 | Cfc | DNF | Ongoing | 2002–2006 | Wang et al. (2005) | Saigusa and Wang (2025) |
| JP-Sb1 | | Japan | JP | Sarobetsu Mire Moss | 45.104722 | 141.688194 | 6 | Dfb | WET | Completed | 2007–2010 | Hirano et al. (2016) | Hirano (2025a) |
| JP-Sb2 | | Japan | JP | Sarobetsu Mire Sasa | 45.103611 | 141.680833 | 4 | Dfb | WET | Completed | 2007–2010 | Hirano et al. (2016) | Hirano (2025b) |
| JP-Tef | TSE | Japan | JP | CC-LaG Teshio Experimental Forest | 45.055808 | 142.107122 | 79.47 | Dfb | DNF | Ongoing | 2001–2023 | Takagi et al. (2009) | Takagi and Takahashi (2025) |
| JP-MBF | MBF | Japan | JP | Moshiri Birch Forest Site | 44.38416667 | 142.3186111 | 596 | Af | DBF | Completed | 2003–2011 | Nakai et al. (2006) | Nakai et al. (2025a) |
| JP-MMF | MMF | Japan | JP | Moshiri Mixd Forest Site | 44.32194444 | 142.2613889 | 343 | Af | MF | Completed | 2003–2011 | Nakai et al. (2006) | Nakai et al. (2025b) |
| JP-BBY | BBY | Japan | JP | Bibai bog | 43.32296 | 141.81079 | 17 | Dfb | WET | Completed | 2012–2021 | Ueyama et al. (2020c) | Ueyama et al. (2025e) |
| JP-Km1 | | Japan | JP | Kushiro Mire: Onnenai Fen | 43.107511 | 144.330906 | 4.9 | Dfb | WET | Completed | 1994–1996, 1998 | Miyata et al. (1997) | Harazono and Miyata (2025a) |
| JP-Km2 | | Japan | JP | Kushiro Mire: Akanuma Bog | 43.1 | 144.35 | 7 | Dfb | WET | Completed | 1998–1999 | Miyata et al. (2001) | Harazono and Miyata (2025b) |
| JP-Spp | SAP | Japan | JP | Sapporo forest meteorology research site | 42.9868431 | 141.3853305 | 174 | Dfb | DBF | Ongoing | 2000–2018 | Yamanoi et al. (2015) | Mizoguchi and Kitamura (2025) |
| CN-In4 | | China | CN | Inner Mongolia maize | 42.94413333 | 120.7266222 | 354 | Bsk | CRO | Completed | 1994 | Li et al. (2000) | Harazono and Takagi (2025d) |
| CN-In5 | | China | CN | Inner Mongolia no grazing | 42.93415833 | 120.7090778 | 355 | Bsk | GRA | Completed | 1992–1994 | Li et al. (2000) | Harazono and Takagi (2025e) |
| CN-In6 | | China | CN | Inner Mongolia heavy grazing | 42.93401389 | 120.7115472 | 355 | Bsk | GRA | Completed | 1992–1994 | Li et al. (2000) | Harazono and Takagi (2025f) |
| CN-In8 | | China | CN | Inner Mongolia medium grazing | 42.93396667 | 120.7105306 | 355 | Bsk | GRA | Completed | 1992, 1994 | Li et al. (2000) | Harazono and Takagi (2025h) |
| CN-In2 | | China | CN | Inner Mongolia grassland | 42.93396389 | 120.7109639 | 355 | Bsk | GRA | Completed | 1991 | Li et al. (2000) | Harazono and Takagi (2025b) |
| CN-In7 | | China | CN | Inner Mongolia light grazing | 42.93391944 | 120.7096056 | 355 | Bsk | GRA | Completed | 1992–1994 | Li et al. (2000) | Harazono and Takagi (2025g) |
| CN-In1 | | China | CN | Inner Mongolia dune | 42.92970833 | 120.70735 | 356 | Bsk | BSV | Completed | 1990–1991 | Li et al. (2000) | Harazono and Takagi (2025a) |
| CN-In3 | | China | CN | Inner Mongolia soybean | 42.94413333 | 120.7266222 | 354 | Bsk | CRO | Completed | 1994 | Li et al. (2000) | Harazono and Takagi (2025c) |

| Site code (BADM) | AsiaFlux ID | Country | Country ID | Site name | Latitude (degree) | Longitude (degree) | Elevation (m) | Köppen climate | IGBP (land use) | Status | Years | Reference | Data citation |
|---|---|---|---|---|---|---|---|---|---|---|---|---|---|
| JP-Tmk | TMK | Japan | JP | Tomakomai Flux Research Site | 42.736972 | 141.516944 | 140 | Dfb | DNF | Completed | 2001–2003 | Hirano et al. (2003) | Hirata and Hirano (2025) |
| JP-Tmd | TMK | Japan | JP | Tomakomai Flux Research Site Disturbed | 42.735911 | 141.523147 | 117 | Dfb | DBF | Ongoing | 2005–2023 | Hirano et al. (2017) | Hirano and Hirata (2025) |
| JP-Toc | | Japan | JP | Tomakomai Crane site | 42.709727 | 141.565898 | 96 | Dfb | DBF | Ongoing | 2010–2014 | Nakamura et al. (2014) | Nakaji et al. (2025) |
| JP-Tom | TOE | Japan | JP | Tomakomai Experimental Forest | 42.698906 | 141.571488 | 90 | Dfb | DBF | Completed | 1999–2013 | Shibata et al. (2005) | Nakaji (2025) |
| JP-Srk | SRK | Japan | JP | Shirakami Beech Forest Site | 40.565485 | 140.127794 | 340 | Dfa | DBF | Ongoing | 2010–2016 | Ishida et al. (2009) | Ishida (2025) |
| JP-Api | API | Japan | JP | Appi forest meteorology research site | 40.0013585815243 | 140.936585918296 | 831 | Dfa | DBF | Ongoing | 2000–2022 | Yasuda et al. (2012) | Yasuda (2025a) |
| JP-Mra | MRA | Japan | JP | Muramatsu Agricultural Field | 37.690275 | 139.194429 | 43 | Cfa | CRO | Ongoing | 2023 | Boiarskii and Hasegawa (2019) | Nagano and Hasegawa (2025) |
| CN-HaM | QHB | China | CN | Qinghai Flux Research Site | 37.607432 | 101.332 | 3250 | BSk | GRA | Ongoing | 2001–2014 | Du et al. (2021) | Du et al. (2025) |
| JP-NsM | NSS | Japan | JP | Nasu Research Station, Manure Application Plot | 36.91583333 | 139.9358333 | 320 | Cfa | GRA | Completed | 2004–2015 | Matsuura et al. (2023) | Matsuura (2025a) |
| JP-NsC | NSS | Japan | JP | Nasu Research Station, Chemical Fertilizer Plot | 36.915 | 139.9366667 | 320 | Cfa | GRA | Completed | 2004–2015 | Matsuura et al. (2023) | Matsuura (2025b) |
| JP-Kzw | KZW | Japan | JP | Karuizawa | 36.406667 | 138.5725 | 1385 | Dfb | DBF | Completed | 2001–2008 | Nakaya et al. (2006) | Nakaya et al. (2025) |
| JP-Tkb | | Japan | JP | Tsukuba Experimental Watershed | 36.173379 | 140.176634 | 341 | Cfa | ENF | Ongoing | 2014, 2018–2021 | Iida et al. (2020) | Shimizu et al. (2025b) |
| JP-Tak | TKY | Japan | JP | Takayama deciduous broadleaf forest site | 36.14616667 | 137.4231111 | 1425 | Dfb | DBF | Ongoing | 1998–2021 | Murayama et al. (2024a) | Murayama et al. (2025b) |
| JP-Ta2 | TKC | Japan | JP | Takayama evergreen coniferous forest site | 36.139722 | 137.370833 | 800 | Dfb | ENF | Ongoing | 2005–2022 | Saitoh et al. (2010) | Saitoh and Tamagawa (2025) |
| JP-Tgf | TGF | Japan | JP | Terrestrial Environment Research Center, University of Tsukuba | 36.11353 | 140.09488 | 27 | Cfa | GRA | Completed | 2002–2022 | Shimoda et al. (2005) | Asanuma and Shimoda (2025) |
| JP-KaP | | Japan | | Kasumigaura lotus paddy | 36.08 | 140.24 | 3 | Cfa | CRO | Completed | 1997–1998 | Takagi et al. (2003) | Harazono et al. (2025) |
| JP-Mse | MSE | Japan | JP | Mase paddy flux site | 36.05393 | 140.02693 | 11 | Cfa | CRO | Ongoing | 2001–2009 | Saito et al. (2005) | Ono (2025) |
| JP-SwL | SWL | Japan | JP | Suwa Lake Site | 36.04657222 | 138.1083528 | 758 | Dfc | WAT | Ongoing | 2015–2023 | Iwata et al. (2018) | Iwata (2025b) |
| JP-KaL | | Japan | JP | Koshin, Lake Kasumigaura | 36.037778 | 140.404167 | 0.26 (at the water level of Y.P.1.1 m) | Cfa | WAT | Ongoing | 2007–2022 | Sugita et al. (2020) | Sugita (2025) |
| JP-Nsb | | Japan | JP | NIAES Soybean | 36.024303 | 140.114975 | 24 | Cfa | CRO | Completed | 1990 | Harazono et al. (1992) | Harazono (2025a) |
| JP-Yrp | | Japan | JP | Yawara Rice paddy | 36.00766667 | 140.0301752 | 23 | Cfa | CRO | Completed | 1993–1995 | NA | Harazono (2025b) |
| JP-Kwg | KWG | Japan | JP | Kawagoe forest meteorology research site | 35.8725 | 139.4869 | 41 | Cfa | DBF | Completed | 1997–2002 | Yasuda et al. (1998) | Yasuda (2025b) |
| JP-Shn | | Japan | JP | Shinshu University Experimental Forest Site | 35.865755 | 137.932563 | 775 | Dfa | MF | Ongoing | 2014–2019 | NA | Iwata and Suzuki (2025) |
| JP-Nkm | NKM | Japan | JP | Nishikoma Site | 35.808064 | 137.833883 | 2641 | Dfb | ENF | Ongoing | 2018–2023 | NA | Iwata (2025a) |
| JP-Fmt | | Japan | JP | Field Museum Tama Hills | 35.638745 | 139.379748 | 168 | Cfa | MF | Ongoing | 2013–2023 | Matsuda et al. (2015) | Takagi and Matsuda (2025) |
| JP-Kgu | | Japan | JP | Kugahara urban residential area | 35.582859 | 139.693543 | 18.5 | Cfa | URB | Completed | 2001–2002 | Moriwaki and Kanda (2004) | Kanda and Moriwaki (2025) |
| JP-Fjy | FJY | Japan | JP | Fujiyoshida forest meteorology research site | 35.45454 | 138.76225 | 1043 | Cfa | ENF | Ongoing | 2000–2021 | Mizoguchi et al. (2012) | Takanashi et al. (2025a) |

| Site code (BADM) | AsiaFlux ID | Country | Country ID | Site name | Latitude (degree) | Longitude (degree) | Elevation (m) | Köppen climate | IGBP (land use) | Status | Years | Reference | Data citation |
|---|---|---|---|---|---|---|---|---|---|---|---|---|---|
| JP-Fhk | FHK | Japan | JP | Fuji Hokuroku Flux Observation Site | 35.44355577 | 138.7646931 | 1100 | Cfa | DNF | Ongoing | 2006–2023 | Takahashi et al. (2015) | Takahashi et al. (2025) |
| JP-Hrt | | Japan | JP | Hiratsuka Rice Paddy | 35.362778 | 139.338056 | 6.98 | Cfa | CRO | Completed | 2013 | Komiya (2015) | Komiya (2025a) |
| JP-SMF | SMF | Japan | JP | Seto Mixed Forest Site | 35.261528 | 137.07875 | 212 | Cfa | MF | Completed | 2002–2016 | Matsumoto et al. (2008) | Kotani and Ohta (2025) |
| JP-Nuf | | Japan | JP | Nagoya University Forest | 35.15241667 | 136.9718889 | 66 | Cfa | DBF | Completed | 2000–2001 | Hiyama et al. (2005) | Awal and Ohta (2025a) |
| JP-Tdf | | Japan | JP | Toyota Deciduous Forest | 35.03588889 | 137.1857778 | 104 | Cfa | DBF | Completed | 2002–2004 | Awal et al. (2010) | Awal and Ohta (2025b) |
| JP-Yms | YMS | Japan | JP | Yamashiro forest meteorology research site | 34.790278 | 135.840939 | 220 | Cfa | DBF | Ongoing | 2000–2023 | Kominami et al. (2008) | Takanashi et al. (2025b) |
| JP-Nap | | Japan | JP | Nunoike Agricultural Pond | 34.77485 | 134.892442 | 40 | Cfa | WAT | Completed | 2021–2023 | NA | Sakabe and Itoh (2025) |
| JP-Ako | AKO | Japan | JP | Akou green belt | 34.735192 | 134.374798 | 10.5 | Cfa | EBF | Completed | 2000–2003 | Kosugi et al. (2005) | Kosugi and Takanashi (2025) |
| JP-Sac | SAC | Japan | JP | Sakai City Office | 34.57391389 | 135.4828889 | 17 | Cfa | URB | Ongoing | 2008–2023 | Ueyama and Takano (2022) | Ueyama (2025d) |
| JP-Ozm | IZM | Japan | JP | Oizumi Urban Park | 34.563469 | 135.533483 | 22 | Cfa | URB | Completed | 2015–2016 | Ueyama and Ando (2016) | Ueyama (2025a) |
| JP-Om1 | OM1 | Japan | JP | B11 building in Osaka Metropolitan University | 34.547177 | 135.502861 | 27 | Cfa | URB | Ongoing | 2014–2023 | Ueyama and Ando (2016) | Ueyama (2025b) |
| JP-Om2 | OM2 | Japan | JP | Farm field in Osaka Metropolitan University | 34.542452 | 135.508227 | 50 | Cfa | GRA | Ongoing | 2022–2023 | NA | Ueyama (2025c) |
| JP-Hc3 | | Japan | JP | Hachihama Experimental Farm: Double Crop | 34.539672 | 133.911731 | −0.25 | Cfa | CRO | Completed | 2005–2009 | Takimoto et al. (2010) | Takimoto and Iwata (2025b) |
| JP-Hc1 | | Japan | JP | Hachihama Experimental Farm: the International Rice Experiment | 34.53789167 | 133.9267972 | 0 | Cfa | CRO | Completed | 1996 | Harazono et al. (1998) | Harazono (2025c) |
| JP-Hc2 | HCH | Japan | JP | Hachihama Experimental Farm | 34.537518 | 133.927545 | −1 | Cfa | CRO | Completed | 1999–2008 | Ohtaki (1984) | Takimoto and Iwata (2025a) |
| JP-Khw | KHW | Japan | JP | Kahoku Experiment watershed | 33.13658 | 130.70834 | 196 | Cfa | ENF | Ongoing | 2000–2003, 2007–2021 | Shimizu et al. (2015) | Kitamura et al. (2025) |
| JP-Ynf | YNF | Japan | JP | Yona-Field Tower Site | 26.751 | 128.212667 | 213 | Cfa | EBF | Ongoing | 2013–2022 | Matsumoto et al. (2023) | Matsumoto et al. (2025) |
| TH-Kog | | Thailand | TH | Kog-Ma Watershed | 18.8 | 98.9 | 1265 | Af | EBF | Completed | 2005–2013 | Kume et al. (2007) | Kumagai and Takamura (2025a) |
| TH-Mae | | Thailand | TH | Mae Moh plantation | 18.38333333 | 99.71666667 | 380 | Aw | DBF | Completed | 2005–2016 | Igarashi et al. (2015) | Kumagai and Takamura (2025b) |
| TH-Kms | | Thailand | TH | Kamphaeng Saen Rice Paddy | 14.009167 | 99.984167 | 4.74 | Aw | CRO | Completed | 2014 | Komiya (2015) | Komiya (2025b) |
| KH-Kmp | | Cambodia | KH | Kampong Thom Lowland Dry Evergreen Forest | 12.74457978 | 105.4785661 | 95 | Am | EBF | Ongoing | 2011–2014 | Kabeya et al. (2021) | Shimizu et al. (2025a) |
| MY-LHP | LHP | Malaysia | MY | Lambir Hills National Park | 4.201007 | 114.039079 | 140 | Af | EBF | Completed | 2009–2019 | Takamura et al. (2023) | Kumagai et al. (2025) |
| ID-Pag | | Indonesia | ID | Palangkaraya Undrained Forest | −2.323916667 | 113.9043917 | 22 | Am | EBF | Ongoing | 2004–2019 | Hirano et al. (2024) | Hirano and Ohkubo (2025c) |
| ID-PaB | | Indonesia | ID | Palangkaraya Drained Burnt forest | −2.340796 | 114.0379 | 14 | Am | OSH | Completed | 2004–2017 | Ohkubo et al. (2021) | Hirano and Ohkubo (2025a) |
| ID-PaD | PDF | Indonesia | ID | Palangkaraya Drained forest | −2.346070697 | 114.036408 | 26 | Am | EBF | Completed | 2001–2017 | Hirano et al. (2024) | Hirano and Ohkubo (2025b) |

NA: not available.

July 2024) and JapanFlux (https://www.japanflux.org/, last access: 11 July 2024). We also collected information on previous studies that reported micrometeorological measurements from domestic researcher connections and literature surveys. The collected data were from eddy covariance observations that were carried out by site teams affiliated with Japanese research institutes and universities. By this criterion, the dataset covers not only Japan but also other countries, such as Russia, China, Mongolia, Cambodia, Thailand, Malaysia, and Indonesia. Most of the sites were established for long-term monitoring of $CO_2$ fluxes, but intensive observations for about a week in the 1990s were also included in the dataset. Because the data format differed for each team, we reformatted the file to the FLUXNET format (https://ameriflux.lbl.gov/data/aboutdata/data-variables/, last access: 11 July 2024) after consultation with each site team. Generally, non-gap-filled data were provided by the site teams, but some teams provided gap-filled meteorological and flux data in addition to the non-gap-filled data. The JapanFlux2024 dataset differs from datasets such as FLUXNET2015 in that it provides site principal investigators (PIs), with increased flexibility in data screening. When clear anomalies were identified, quality control procedures were applied by the management team in collaboration with the respective site PI.

The dataset consists of data from 83 sites with 683 site-years, of which 52 sites are located in Japan (Fig. 2; Table 1). The dataset includes 43 forest sites, 15 grassland sites, 5 wetland sites, 10 cropland sites, 3 lake and pond sites, and 4 sites in urban landscapes. Sites that suffered from various types of disturbance are also included: wind damage by typhoon (JP-Tmd, JP-Spp), fire (RU-NeB, ID-PaB), harvesting (RU-NeC, JP-Tef), thinning (JP-Fhk), insect outbreak (JP-Api), drainage (ID-Pag), and mowing (JP-NsC, JP-NsM, JP-Tgf, JP-Om2). The data records started in 1990 at a soybean cropland in Japan (Harazono et al., 1992), increased in number in the early 2000s, and peaked at 34 sites in 2008, 2014, and 2015 (Fig. 3). More recently, the number of data records gradually declined owing to site closure or the fact that the data have not been processed yet. The longest record was 24 years (JP-Tak and JP-Yms; both deciduous broadleaf forests) (Fig. 4). There are 26 sites with observation records of $CO_2$ flux for more than 10 years and six sites with those for more than 20 years (JP-Tef, JP-Tmk/JP-Tmd, JP-Api, JP-Fjy, JP-Tak, JP-Yms). Note that JP-Tmk and JP-Tmd represent a continuous observation series, although they are assigned different site IDs. At 12 sites, data records are available for less than 1 year. Data for $CH_4$ flux are available at six sites (JP-BBY, JP-SwL, JP-Nap, JP-Hrt, JP-Sac, JP-Om1).

## 2.2 Gap-filling meteorological variables

As with the FLUXNET2015 dataset (Pastorello et al., 2020; Vuichard and Papale, 2015), the meteorological variables were filled using the European Center for Medium-Range Weather Forecasts Reanalysis v5 (ERA5) data (Hersbach et al., 2020). Instead of using ERA5, we used the gap-filled meteorology if the site teams had filled the gaps. If meteorological variables for multiple sensors or positions were available, these variables were prioritized and aggregated; if data were missing in the highest-priority dataset, they were filled with values from the second-highest-priority dataset or, if that were also unavailable, based on the priority order. The gaps in the aggregated meteorological variables were then filled with ERA5 data because measured variables were less biased than ERA5, even when measured at different locations within a site. Air temperature, relative humidity, wind speed, downward shortwave radiation, downward longwave radiation, precipitation, and atmospheric pressure were filled using ERA5 after correcting biases at each site for each year. Linear regression for meteorological variables (except precipitation) between observations and ERA5 was determined and then applied to correct site-specific biases in ERA5 to fill the data gaps. Water vapor pressure was calculated from the relative humidity, and the gaps in relative humidity were filled using the gap-filled water vapor pressure and air temperature rather than directly using the relative humidity. If all meteorological variables were missing in some years when constructing the linear regression, the bias was corrected using a regression for the entire multi-year data record. For precipitation (denoting rainfall plus snowfall), we determined the ratio of the annual precipitation between observations and ERA5 during the period when observed precipitation were available and then filled hourly or half-hourly precipitation after multiplying the ERA5-based precipitation by the ratio. If only the rainfall was measured, the correction ratio was determined using liquid precipitation, which was defined as precipitation when the relative humidity was below the critical relative humidity ($RH_{cri}$; %): $RH_{cri} = 92.5 - 7.5T$, where $T$ is the air temperature (Matsuo and Sasyo, 1981).

## 2.3 Gap-filling and flux partitioning

Gap-filling and flux partitioning were conducted based on REddyProc (version 1.3.2; Wutzler et al., 2018). First, the friction velocity ($u^*$) threshold was determined for the identified low-turbulence conditions during the nighttime using the moving-point method (Papale et al., 2006). The $u^*$ threshold was determined from the temperature sensitivity of nighttime net ecosystem exchange (NEE) by seasonal clustering, an approach that is widely used in the FLUXNET community. In the moving point method, the $u^*$ threshold was first determined for each of the four seasons, and the maximum value among them was used for the entire year. Thus, the determined $u^*$ threshold was conservative (Papale et al., 2006). In this dataset, we determined the $u^*$ threshold for each year to consider its potential shift over the years, which is termed as a variable $u^*$ threshold (vUT). The vUT differs slightly from the definition of the variable $u^*$ threshold (VUT) in FLUXNET2015 (Pastorello et al., 2020) (Table 2):

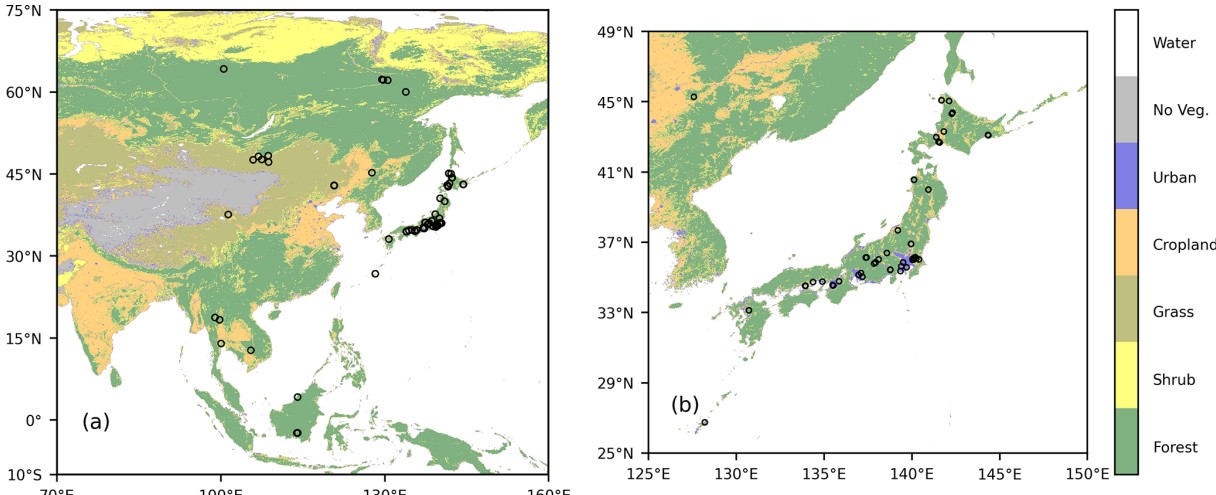

**Figure 2.** Distributions of the sites that constitute the JapanFlux2024 database on a land cover map provided by the MOD12 product (version 6.1; Sulla-Menashe et al., 2019): a map of the Asia region **(a)** and an enlarged map showing Japan **(b)**.

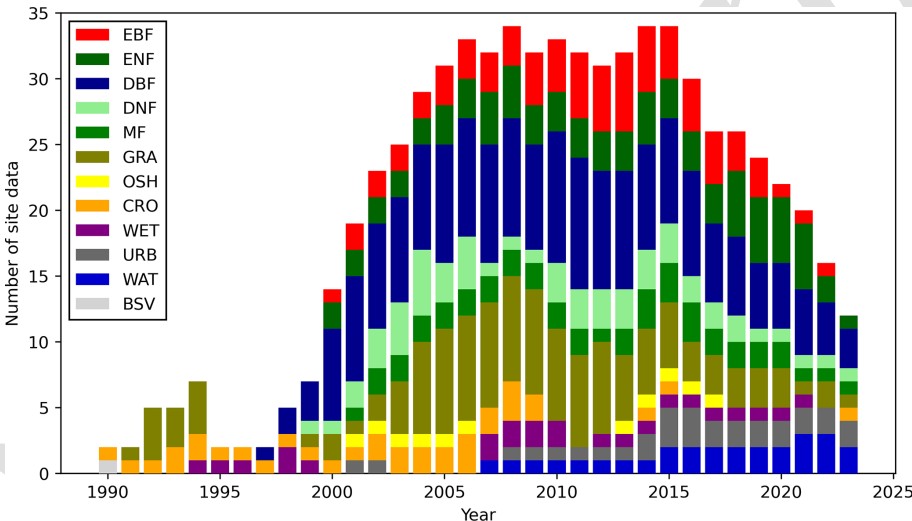

**Figure 3.** Number of site data records for each year. Land cover types: evergreen broadleaf forest (EBF), evergreen needleleaf forest (ENF), deciduous broadleaf forest (DBF), deciduous needleleaf forest (DNF), mixed forest (MF), grassland (GRA), open shrubland (OSH), cropland (CRO), wetland (WET), urban (URB), lake (WAT), and barren sparse vegetation (BSV).

the VUT in FLUXNET2015 was determined by pooling data from each year along with data from the immediately preceding and following years (if available). The $u^*$ threshold was determined with 100 bootstrap replicates, where reference (original data obtained without using a bootstrapped sample) and the 5th, 50th, and 95th percentiles of the estimated $u^*$ threshold were used for subsequent data filtering, gap-filling, and flux partitioning. Here, the nighttime was defined as downward shortwave radiation $< 10\,\mathrm{W\,m^{-2}}$ and was further confirmed using exact solar time at the site location. On the basis of the estimated $u^*$ threshold, nighttime $CO_2$ fluxes and/or NEE were eliminated. This dataset does not include the estimation of NEE using the constant $u^*$ threshold (CUT) nor the advanced uncertainty estimation provided with the _REF suffix, as implemented in the ONE-FLUX pipeline (Pastorello et al., 2020). For urban sites, the threshold was generally not used for two reasons (e.g., Liu et al., 2012; Ueyama and Ando, 2016): (1) nighttime $CO_2$ fluxes were not expected to correlate with air temperature, making it difficult to evaluate the correct $u^*$ threshold, and (2) the surface layer was often unstable, even at night. Consequently, the $u^*$ filtering was not applied for highly urbanized sites (JP-Sac and JP-Kgu).

Gaps in sensible heat flux ($H$), latent heat flux ($LE$), and NEE were filled using marginal distribution sampling (MDS) based on REddyProc. In MDS, a lookup table (LUT) with

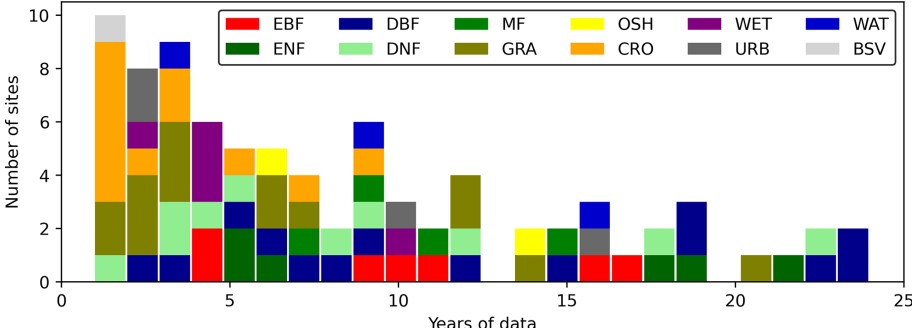

**Figure 4.** Number of site records for different durations of data records. Sites affected by a disturbance that changed the vegetation type during the observational period were classified according to the dominant land cover type: JP-Tef as DNF, JP-Tmd as DBF, and ID-PaB as OSH. Land cover type abbreviations are as in Fig. 3.

air temperature, downward shortwave radiation, and vapor pressure deficit (VPD) was created for a 7 d window. When data gaps could not be filled with this window, they were filled in the following order: (1) LUT was applied with a 14 d window, (2) the mean diurnal variation method (Falge et al., 2001) was applied with a 1 or 2 d window, and (3) LUT was applied with a 21 d window, which was increased with a 7 d step until reaching a 70 d window if not enough data points were available. NEE were filled using MDS with the four different $u^*$ thresholds (reference, 5th, 50th, and 90th percentiles values), whereas $H$ and $LE$ were filled without applying the $u^*$ threshold. In addition to the fluxes, net radiation, soil temperature, ground heat flux, and photosynthetic photon flux density (PPFD) were also filled using MDS. In the data collected from the site teams, energy imbalance correction (Twine et al., 2000) was not applied for $H$ and $LE$ at any sites; thus, the gap-filled $H$ and $LE$ were not corrected for the energy balance closure.

Using REddyProc, NEE was partitioned into gross primary productivity (GPP) and ecosystem respiration (RECO) using two methods: nighttime flux partitioning and daytime flux partitioning. In the nighttime partitioning method, nighttime NEE was parameterized on the basis of the temperature response function (Lloyd and Taylor, 1994) with a 7 d window, and then this function was used to calculate daytime and nighttime RECO. GPP was determined by subtracting RECO from NEE. In the daytime partitioning method (Lasslop et al., 2010), the common rectangular hyperbolic light-response curve was determined with a 4 d window, where the function accounted for the VPD effect on the initial slope of the light-response curve and the temperature effect of respiration. GPP and RECO using the daytime partitioning method were calculated based on a fitted model that combines a light-response curve and a temperature-dependent respiration model; thus, the daytime method did not directly add up the observed NEE (Wutzler et al., 2018). Using the two methods, fluxes were partitioned for NEE with different $u^*$ thresholds.

## 2.4 Site-specific considerations

For the sites with heterogeneous land surfaces – JP-Khw and JP-Ozm – the dominant land surface fluxes were extracted using wind sectors. JP-Khw is an evergreen needleleaf plantation forest consisting of *Cryptomeria japonica* (sugi) and *Chamaecyparis obtusa* (hinoki), but evergreen or deciduous broadleaf trees grow in gaps in some wind sectors. The $H$, $LE$, and $CO_2$ fluxes for sugi, which occupies the dominant wind sector area (the right-bank side), were extracted as quality control with footprint, "_QC_FP" (Table 2). To extract these flux data, daytime fluxes for a wind sector on the right bank were selected, but nighttime fluxes for all wind sectors were used to increase data availability because there were no clear differences in nighttime fluxes among wind sectors. Gap-filling and flux partitioning were done only for the extracted data. JP-Ozm is located at the edge of an urban park; thus, measured flux representing this park (Ueyama and Ando, 2016) were selected and designated "_QC_FP" in addition to the variables for measured fluxes representing both sectors of the urban park and other land covers. Gap-filling and flux partitioning were done only for the extracted data, which represented the urban park. These extracted flux data ("_QC_FP") were included in the ALLVARS files (described in Sect. 2.5) in addition to measured fluxes for all sectors, and the gap-filled extracted fluxes were included in the CORE-VARS files (described in Sect. 2.5).

Flux partitioning and gap-filling for JP-Nkm, located on a complex mountainous terrain, were conducted using slope-normal shortwave radiation instead of downward shortwave radiation. Horizontally observed incident shortwave radiation was converted to radiation normal to the slope on the basis of the tilt and azimuth angles of the slope and the solar altitude and azimuth angles (Hammerle et al., 2007; Nie et al., 1992) as follows. Horizontally observed incident shortwave radiation was partitioned into direct and diffuse components using the observed diffuse fraction (BF5, Delta-T Devices, UK), and the direct component was converted to that normal to the slope surface. The diffuse component was

**Table 2.** List of variable base names not used in FLUXNET2015, along with their descriptions and related sites.

| Base name | Description | Sites |
|---|---|---|
| Preprocessing variables | | |
| SW_IN_SLOPE_PI | Slope-normal incoming shortwave radiation | JP-Nkm |
| NETRAD_SLOPE_PI | Slope-normal net radiation | JP-Nkm |
| USTAR_QC_FP | Friction velocity qualified with footprint | JP-Ozm |
| H_QC_FP | Sensible heat flux qualified with footprint | JP-Ozm, JP-Khw |
| LE_QC_FP | Latent heat flux qualified with footprint | JP-Ozm, JP-Khw |
| FC_QC_FP | $CO_2$ flux qualified with footprint | JP-Ozm, JP-Khw, JP-Sac |
| NEE_QC_FP | NEE flux qualified with footprint | JP-Khw |
| Post-processing variables | | |
| TA_multiple | Air temperature by multiple sensors or positions | CN-In1, CN-In2, CN-In3, CN-In4, CN-In5, CN-In6, CN-In7, CN-In8, CN-Lsh, JP-BBY, JP-Fhk, JP-Fjy, JP-Hc1, JP-Ozm, JP-Khw, JP-KaP, JP-Km1, JP-Km2, JP-Kzw, JP-MBF, JP-MMF, JP-Nsb, JP-Nuf, JP-Spp, JP-Tgf, JP-Tak, JP-Tmk, JP-Tef, JP-Ynf, JP-Yrp, MN-Udg, MY-LHP, RU-Ege, RU-SkP, RU-Sk2, TH-Kog, TH-Mae, RU-USk |
| RH_multiple | Relative humidity by multiple sensors or positions | CN-In1, CN-In2, CN-In3, CN-In4, CN-In5, CN-In6, CN-In7, CN-In8, CN-Lsh, JP-BBY, JP-Fhk, JP-Fjy, JP-Hc1, JP-KaP, JP-Km1, JP-Km2, JP-Kzw, JP-MBF, JP-MMF, JP-Nsb, JP-Nuf, JP-Spp, JP-Tgf, JP-Tak, JP-Tmk, JP-Tef, JP-Ynf, JP-Yrp, MN-Udg, MY-LHP, RU-Ege, RU-SkP, RU-Sk2, TH-Mae, RU-USk |
| SW_IN_multiple | Incoming shortwave radiation by multiple sensors or positions | JP-Fhk, JP-Spp, JP-Tgf, JP-Tmk, JP-Tef, JP-Ynf |
| P_multiple | Precipitation by multiple sensors or positions | JP-BBY, JP-Khw |
| WS_IN_multiple | Wind speed by multiple sensors or positions | CN-In1, CN-In2, CN-In3, CN-In4, CN-In5, CN-In6, CN-In7, CN-In8, CN-HaM, JP-Fhk, JP-Hc1, JP-KaP, JP-Km1, JP-Km2, JP-Kzw, JP-MBF, JP-MMF, JP-Nsb, JP-Spp, JP-SMF, JP-Tgf, JP-Tef, JP-Yms, JP-Ynf, JP-Yrp, MN-Udg, TH-Kog, TH-Mae, RU-USk |
| G_multiple | Ground heat flux by multiple sensors or positions | JP-Sac, JP-Spp, JP-Ynf, MN-Udg |
| NETRAD_F_MDS | Net radiation filled with MDS | JP-Tef |
| PPFD_IN_F_MDS | PPFD filled with MDS | CN-Lsh, CN-HaM, JP-Km2, JP-MBF, JP-MMF, JP-Nkm, JP-Tgf, RU-SkP, TH-Kog |
| NEE_vUT | Gap-filled NEE with the variable $u^*$ threshold | ALL sites |
| RECO_NT_vUT | RECO with the variable $u^*$ threshold based on the nighttime approach | ALL sites |
| GPP_NT_vUT | GPP with the variable $u^*$ threshold based on the nighttime approach | ALL sites |
| RECO_DT_vUT | RECO with the variable $u^*$ threshold based on the daytime approach | ALL sites |
| GPP_DT_vUT | GPP with the variable $u^*$ threshold based on the daytime approach | ALL sites |

assumed to be isotropic. The total incident shortwave radiation normal to the slope surface was calculated as the sum of the direct component converted as above and the original diffuse component. When the diffuse fraction was not observed, it was estimated from the relationship between the diffuse fraction and cloudiness; the latter was defined as the ratio of observed incident shortwave radiation to extraterrestrial radiation (Wang et al., 2018). The slope-normal shortwave radiation was included as a variable, SW_IN_SLOPE_ PI_ 1_ 1_1, in ALLVARS.

For tropical ecosystems (TH-Kms, TH-Kog, TH-Mae, ML-LHP, ID-PaB), nighttime-based flux partitioning failed because little seasonality in temperature hampered the determination of a significant relationship between nighttime $CO_2$ flux and temperature. For these sites, only daytime partitioning was provided in the dataset. In a subtropical forest (KH-Kmp), the determination of the $u^*$ threshold failed; thus, the $u^*$ threshold was estimated using gap-filled $u^*$ by the site team instead of measured $u^*$ with data gaps. Because the data quality of $u^*$ for KH-Kmp seemed reasonable, we were unable to find out why REddyProc failed to determine the $u^*$ thresholds with measured $u^*$ in KH-Kmp. The $u^*$ threshold for ID-Pag and ID-PaD could not be determined for several years; hence, constant $u^*$ thresholds across these years were determined with REddyProc and applied for the subsequent data processing.

Low availability of nighttime data due to the limited fetch in JP-Ako (Kosugi et al., 2005) hampered determination of the $u^*$ threshold, gap-filling flux for $CO_2$ flux, and flux partitioning with REddyProc. Consequently, no aggregated fluxes longer than half-hourly data for $CO_2$ flux, GPP, and RECO were provided in the dataset.

Fluxes were not partitioned for lakes and a pond (JP-SwL, JP-KaL, JP-Nap) and for an urban center (JP-Sac). For the lakes and pond, gap-filling $H$, $LE$, and $CO_2$ flux was based on MDS. For JP-Sac, gap-filling for $H$ and $LE$ was also based on MDS, but MDS was not applied to $CO_2$ flux because it was controlled by traffic volume and air temperature (Ueyama and Takano, 2022 TS3). Gap-filling for $CO_2$ flux at JP-Sac was conducted by the site team on the basis of random forest regression (Ueyama and Takano, 2022) and was included as FCO2_F_PI in COREVARS and ALLVARS. The $u^*$ threshold was not applied for JP-Nap because the moving point method (Papale et al., 2006) developed for terrestrial ecosystems was not applicable to the pond.

In this dataset, $CH_4$ fluxes were not gap-filled because (1) consistent gap-filling was not possible because of missing important variables, such as water table depth, and (2) inconsistent processes control $CH_4$ emissions on different land surfaces, such as a rice paddy (JP-Hrt), bog (JP-BBY; Ueyama et al., 2020c, b), lake (JP-SwL; Iwata et al., 2018), pond (JP-Nap), and urban landscapes (JP-Sac, JP-Om1; Takano and Ueyama, 2021). If the gap-filled $CH_4$ fluxes were provided by the site team (i.e., JP-BBY), the data were included as FCH4_F_PI in COREVARS; otherwise, non-gap-filled data were included in ALLVARS.

## 2.5 Data format

The dataset was prepared in a format partially compatible with the FLUXNET format, although the content and split of variables between ALLVARS and COREVARS were slightly different from FLUXNET2015 (Pastorello et al., 2020) (Table 2), which consists of files separated by sites, temporal aggregation (i.e., half-hourly/hourly, daily, weekly, monthly, and annual), and the data product, i.e., ALLVARS and COREVARS, as described later. The separated files for ALLVARS and COREVARS were combined into two zip files for each site.

The following file naming rules (Pastorello et al., 2020) were followed: [SITE_ID]_JapanFlux2024_[DATA_PRODUCT]_ [RESOLUTION]_[FIRST_YEAR]- [LAST_YEAR]_[SITE_ VERSION]-[CODE_VERSION].csv.

[SITE_ID] is the site ID. For the CC-SSS format, CC is a two-letter country code, and SSS is the three-character site code. [Data_PRODUCT] represents the data types: ALLVARS, COREVARS, AUXMETEO, AUXNEE, or ERA5. COREVARS is the data type representing selected data variables, including basic micrometeorological data and fluxes, and quality information flags. ALLVARS is a data file representing all variables of data products, including variables listed in COREVARS, original data before the processing pipeline, and internal variables. AUXMETEO includes auxiliary variables related to the meteorological downscaling of ERA5. ERA5 includes the meteorological data from ERA5 for 1990–2024. [RESOLUTION] is the temporal resolution of the data products: HH (half-hourly time step), HR (hourly time step), DD (daily time step), WW (weekly time step), MM (monthly time step), and YY (annual time step). [FIRST_YEAR] is the first year in the file, and [LAST_YEAR] is the last year in the file. The first and last years are based on the years in which the micrometeorological measurements were conducted, except for ERA5, where the first year is 1990 and the last year is 2024 for all sites. [SITE_VERSION] is the version of the original dataset, and [CODE_VERSION] is the code of the data processing pipeline used to process the dataset.

The COREVARS file included variables for basic meteorology and turbulent fluxes. The gap-filled meteorological variables of air temperature, incoming shortwave radiation, incoming longwave radiation, relative humidity, VPD, atmospheric pressure, precipitation, wind speed, net radiation, ground heat flux, soil temperature, PPFD, $CO_2$ concentration, soil water content, and potential shortwave radiation (top of atmosphere) were included. If the original data provided by a site team included wind direction, outgoing shortwave radiation, outgoing longwave radiation, outgoing

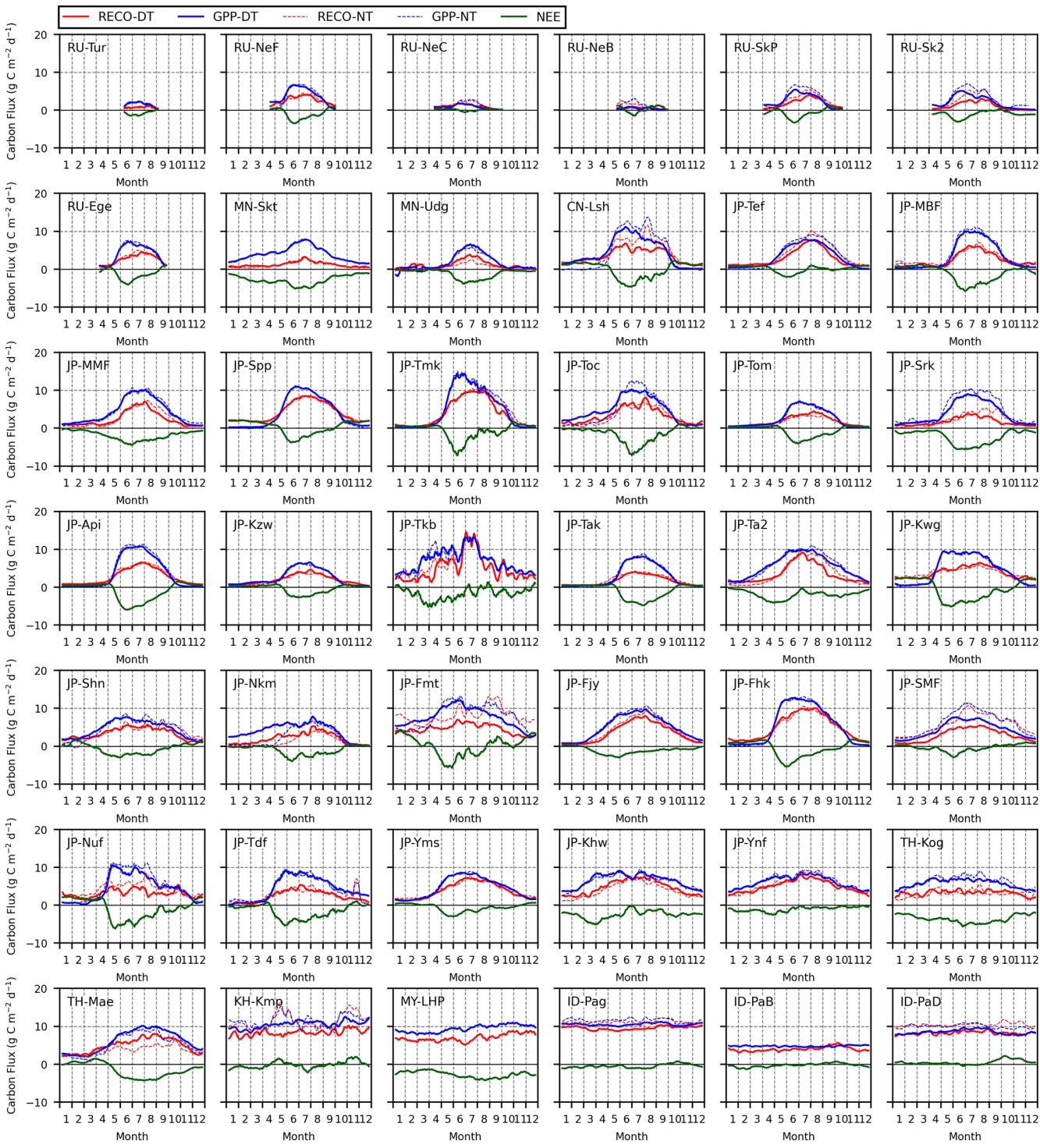

**Figure 5.** Mean seasonality of GPP, RECO, and NEE across forest sites. GPP and RECO were partitioned using the daytime method (DT, solid lines) or the nighttime method (NT, dashed lines). The seasonality is shown when NEE was measured, and those for GPP and RECO are shown when the partitioning was successful. The seasonality is the ensemble mean of the daily fluxes for each day of the year for all years. The sites are ordered according to latitude from high to low. The mean seasonality is shown for sites having data for at least one growing season.

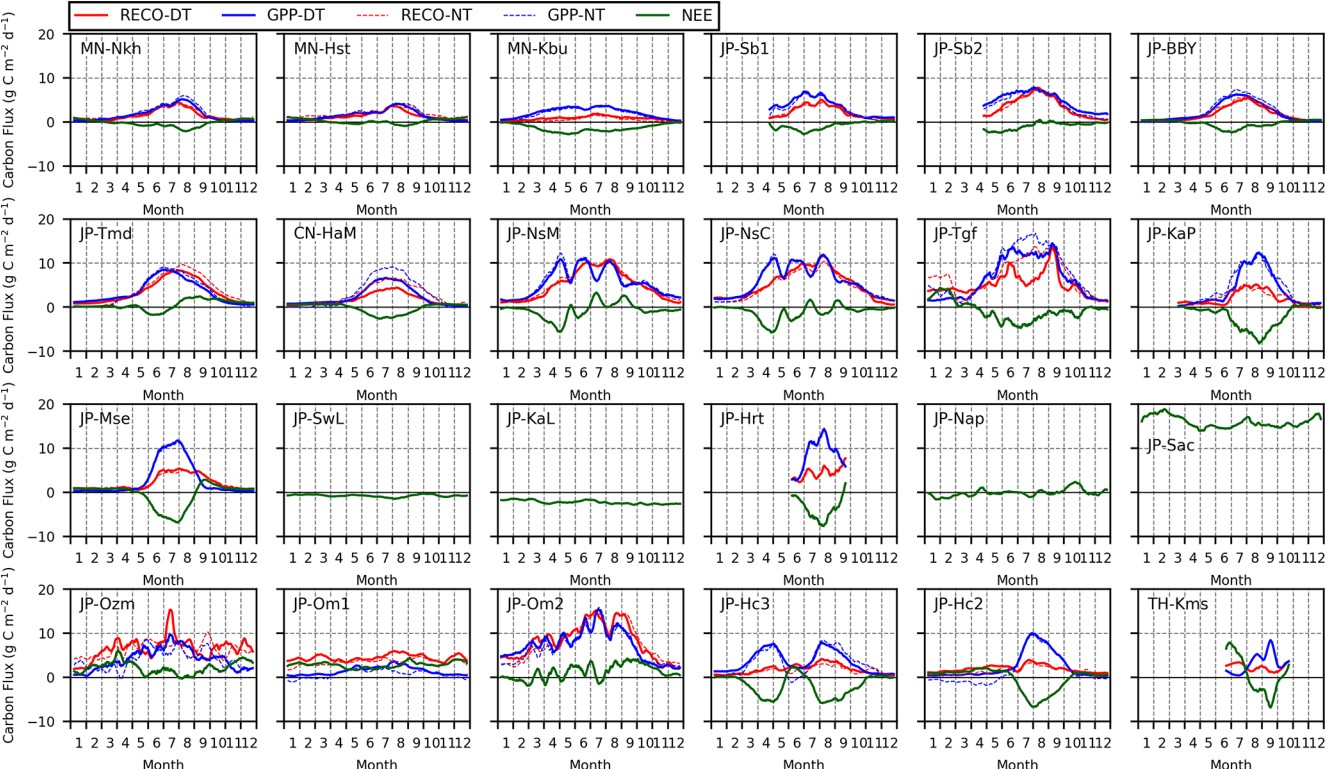

**Figure 6.** Mean seasonality of GPP, RECO, and NEE across sites other than forests. Designations are as in Fig. 5.

PPFD, and $u^*$, non-gap-filled data for these variables were included. Gap-filled soil temperature and soil water content were measured at the shallowest depth, while $CO_2$ concentration was gap-filled for the highest altitude. A quality infor-
5 mation flag was assigned for gap-filled variables, where 0 is the original data, 1 is a gap-filled value of the most reliable quality (calculated using a 14 d window), 2 is a gap-filled value of medium quality (calculated using a 14 to 56 d window), and 3 is the gap-filled value of the least reliable quality
10 (calculated using a window longer than 56 d) (Wutzler et al., 2018). If gap-filled $CH_4$ flux data were provided by the site team (i.e., at JP-BBY), they were included in COREVARS. The COREVARS file was provided with five temporal resolutions (half-hourly/hourly, daily, weekly, monthly, and an-
15 nual aggregations).

The ALLVARS file included the original, unprocessed data, internal variables (aggregated meteorological variables measured at different locations or with different sensors), and meteorological data from ERA5, in addition to the processed
20 variables included in COREVARS. The ALLVARS file is provided with the five temporal resolutions listed above.

For NEE, GPP, and RECO, the unit was $\mu mol\,m^{-2}\,s^{-1}$ for the half-hourly and hourly timescales; $g\,C\,m^{-2}\,d^{-1}$ for the daily, weekly, and monthly timescales; and $g\,C\,m^{-2}\,yr^{-1}$
25 for the annual timescale. For $CH_4$ flux, the unit for the half-hourly and hourly timescales was $nmol\,m^{-2}\,s^{-1}$, whereas the units for the other timescales were the same as those for $CO_2$

fluxes. The units of precipitation were $mm$ for the half-hourly and hourly timescales; $mm\,d^{-1}$ for the daily, weekly, and monthly timescales; and $mm\,yr^{-1}$ for the annual timescale. 30
The units of other variables followed the FLUXNET format (https://ameriflux.lbl.gov/data/aboutdata/data-variables/, last access: 27 December 2024), which did not change with the timescale.

The ERA5 file contains the data for air temperature 35
(TA_ERA5; °C), relative humidity (RH_ERA5; %), VPD (VPD_ERA5; hPa), vapor pressure (e_ERA5; hPa), saturation vapor pressure (e_sat_ERA5; hPa), wind speed (WS_ERA5; $m\,s^{-1}$), atmospheric pressure (PA_ERA5; kPa), incoming shortwave radiation (SW_ERA5; $W\,m^{-2}$), incom- 40
ing longwave radiation (LW_ERA5; $W\,m^{-2}$), and precipitation (P_ERA5; mm). The ERA5 file is provided with the five temporal resolutions listed above. The variables in the ERA5 file were not corrected for the bias in comparison to the site data. 45

Two auxiliary files – for meteorology and the $u^*$ threshold – are provided. The AUXMETEO file includes the following statistics for downscaling ERA5 to the site scale: the linear slope between the measured data and ERA5 (ERA_SLOPE), intercept (ERA_INTERCEPT), root 50
mean square error (ERA_RMSE), and correlation coefficient (ERA_CORRELATION). These statistics are included for each year and for all years when measurements were conducted. The TIMESTAMP column in the AUXMETEO file

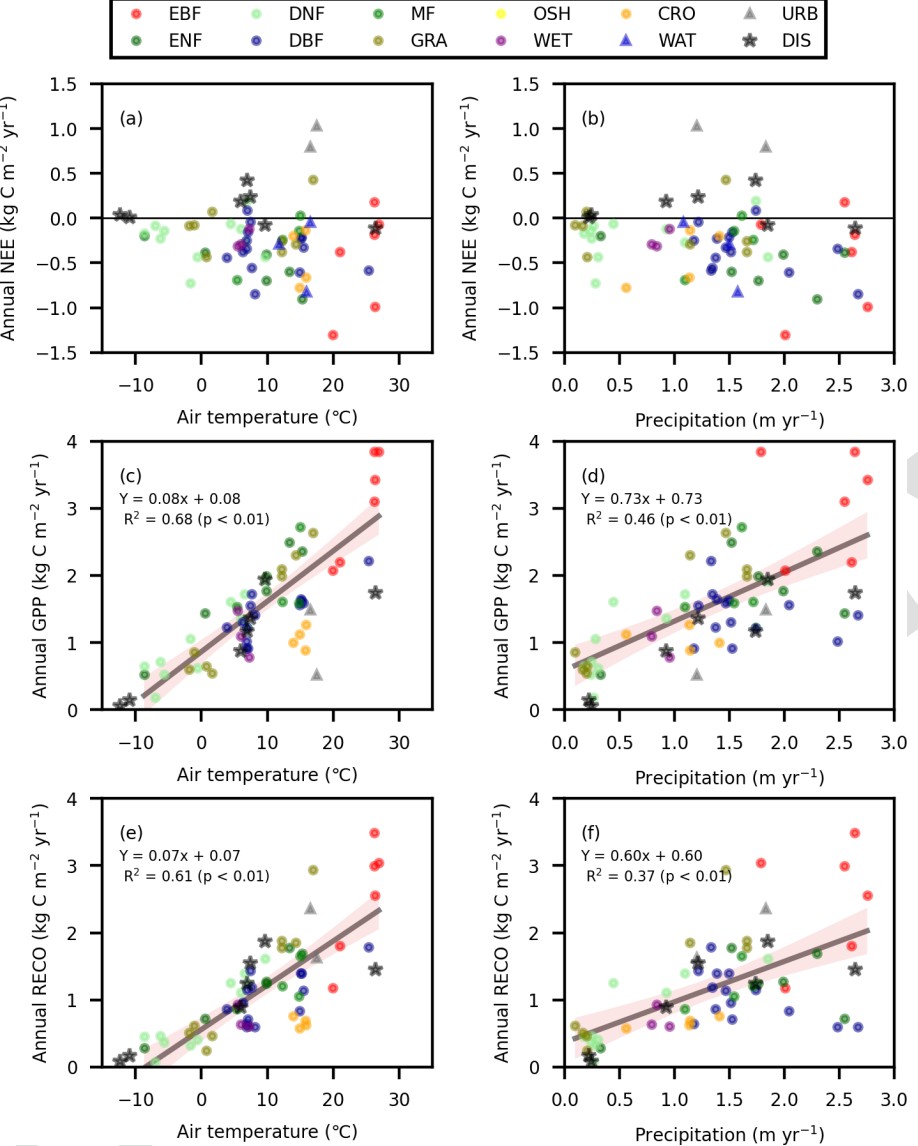

**Figure 7.** Relationships of annual NEE **(a, b)**, GPP **(c, d)**, and RECO **(e, f)** to the mean climate of the annual mean air temperature **(a, c, e)** and annual sum of the precipitation **(b, d, f)**. GPP and RECO were estimated using the daytime method. The stars represent fluxes obtained at disturbed forests, defined as forests that experienced disturbance within the last 10 years. The annual fluxes were calculated based on the sum of the mean seasonality shown in Figs. 5 and 6; missing measurements during the winter at high latitudes were gap-filled as 0. Because the JP-Spp, JP-Tmd, and JP-Tef sites experienced significant disturbance (windthrow or clearcut) during the measurement period, data obtained within 10 years after a disturbance were classified as disturbed forests (DIS). The lines represent linear regressions, with shading showing the confidence intervals ($p < 0.05$), determined by excluding the DIS data. The annual $CO_2$ flux for JP-Sac ($5.8\,\mathrm{kg\,C\,m^{-2}\,yr^{-1}}$) is not shown due to the totally different carbon budget in the urban center compared to those in ecosystems. The values are shown in Table 3. Land cover type abbreviations are in Fig. 3.

represents the year for the statistics, where $-9999$ represents the statistics for the entire year. The AUXNEE file includes the $u^*$ threshold for each year, with the reference threshold and the 5th, 50th, and 95th percentiles of the estimated $u^*$ threshold.

The dataset also includes the BADM files, which are used in the FLUXNET community. Six BADM files are provided:

(1) general information, (2) instrument, (3) instrument operations, (4) vegetation cover, (5) soil, and (6) disturbance and management.

**Table 3.** Summary of mean annual air temperature (TA), annual sum of precipitation (PREC), mean annual downward shortwave radiation (Rsd), mean annual carbon fluxes (NEE, GPP, RECO), mean annual latent heat flux ($LE$), mean annual sensible heat flux ($H$), evapotranspiration (ET), and land cover. The statistics were calculated for the observation years; for disturbed sites, the data were considered separately for the periods before, during, and after disturbance. Disturbed ecosystems were defined as those that experienced disturbance within the last 10 years. GPP, RECO, NEE, $LE$, and ET for boreal forests in Russia that lacked winter measurements (RU-Tur, RU-NeB, RU-NeC, RU-NeF, RU-SkP, RU-Ege, RU-Sk2) were considered 0. GPP, RECO, and NEE at MN-Skt and MN-Kbu were also considered 0 during winter, when the daily mean air temperature was below $-5\,°C$ (indicated by asterisks in the table), to mitigate the influence of the negative values of $CO_2$ fluxes caused by an artifact associated with an open-path sensor. The NA values were listed because missing observations, even after gap-filling fluxes, prevented the calculation of annual fluxes or because the standard flux partitioning was not available for the pond, lakes, and urban landscapes.

| Site ID | Disturbance | Land cover | TA | PREC | Rsd | NEE | GPP | RECO | $LE$ | $H$ | ET |
|---|---|---|---|---|---|---|---|---|---|---|---|
| | | | °C | mm yr$^{-1}$ | W m$^{-2}$ | g C m$^{-2}$ yr$^{-1}$ | g C m$^{-2}$ yr$^{-1}$ | g C m$^{-2}$ yr$^{-1}$ | W m$^{-2}$ | W m$^{-2}$ | mm yr$^{-1}$ |
| RU-Tur | | DNF | −7.0 | 264 | 93 | −83 | 180 | 76 | NA | NA | NA |
| RU-NeB | fire | GRA | −12.3 | 244 | 117 | 37 | 57 | 85 | NA | NA | NA |
| RU-NeF | | OSH | −8.6 | 175 | 117 | −166 | 653 | 455 | 10 | NA | 132 |
| RU-NeC | clearcut | DIS | −10.9 | 224 | 112 | 12 | 147 | 170 | 13 | NA | 162 |
| RU-SkP | | DNF | −5.7 | 238 | 118 | −139 | 523 | 375 | 21 | NA | 265 |
| RU-Sk2 | | ENF | −8.7 | 328 | 117 | −194 | 522 | 283 | 16 | NA | 203 |
| RU-USk | | GRA | −11.6 | 289 | 110 | NA | NA | NA | 15 | NA | 193 |
| RU-Ege | | DNF | −6.3 | 248 | 122 | −225 | 715 | 466 | 20 | NA | 259 |
| MN-Skt | | DNF | −1.7 | 279 | 169 | −722* | 1058* | 324* | 15 | 45 | 189 |
| MN-Udg | | DNF | −0.6 | 303 | 157 | −431 | 617 | 413 | 19 | 29 | 246 |
| MN-Nkh | | GRA | −1.9 | 163 | 171 | −83 | 603 | 516 | 20 | 25 | 253 |
| MN-Hst | | GRA | 1.6 | 197 | 181 | 76 | 541 | 468 | 18 | 27 | 228 |
| MN-Kbu | | GRA | 0.7 | 204 | 183 | −433 | 647* | 248* | 9* | 30 | 116 |
| CN-Lsh | | DNF | 4.4 | 443 | 144 | −60 | 1606 | 1255 | 25 | 36 | 324 |
| JP-Sb1 | | WET | 6.0 | 794 | 145 | −288 | 1098 | 638 | NA | NA | NA |
| JP-Sb2 | | WET | 5.5 | 840 | 140 | −313 | 1476 | 935 | NA | NA | NA |
| JP-Tef | before | DIS | 5.9 | 926 | 125 | 192 | 882 | 898 | 18 | 10 | 232 |
| | clearcut | GRA | 5.9 | 926 | 125 | −118 | 1363 | 1114 | 27 | 13 | 343 |
| | after | MIX | 5.9 | 926 | 125 | −26 | 1238 | 1034 | 21 | 17 | 268 |
| JP-MBF | | DBF | 3.9 | 1373 | 134 | −442 | 1233 | 862 | 37 | 20 | 472 |
| JP-MMF | | MF | 5.4 | 1092 | 134 | −689 | 1537 | 861 | 43 | 21 | 537 |
| JP-BBY | | WET | 7.2 | 953 | 143 | −118 | 785 | 610 | 41 | 12 | 524 |
| JP-Spp | before | DIS | 7.4 | 1215 | 145 | 235 | 1366 | 1554 | 35 | 16 | 444 |
| | windthrow | DBF | 7.4 | 1215 | 145 | −42 | 1557 | 1432 | 40 | 15 | 500 |
| JP-Tmk | | DNF | 6.6 | 1092 | 133 | −270 | 1727 | 1401 | 45 | 31 | 568 |
| JP-Tmd | windthrow | GRA | 7.0 | 1738 | 139 | 421 | 1176 | 1249 | NA | NA | NA |
| | after | DBF | 7.0 | 1738 | 139 | 89 | 1225 | 1147 | 34 | 21 | 427 |
| JP-Toc | | DBF | 7.6 | 1342 | 137 | −556 | 1729 | 1189 | 33 | 30 | 418 |
| JP-Tom | | DBF | 6.9 | 1173 | 128 | −249 | 916 | 642 | NA | 40 | NA |
| JP-Srk | | DBF | 8.1 | 2669 | 129 | −847 | 1408 | 600 | 41 | −1 | 509 |
| JP-Api | | DBF | 6.3 | 1509 | 150 | −375 | 1307 | 958 | 18 | 14 | 235 |
| CN-HaM | | GRA | −1.1 | 97 | 200 | −77 | 862 | 618 | 31 | 23 | 389 |
| JP-NsM | | GRA | 12.2 | 1658 | 150 | −251 | 1989 | 1779 | 55 | 6 | 704 |
| JP-NsC | | GRA | 12.2 | 1658 | 150 | −376 | 2098 | 1880 | 53 | 7 | 674 |
| JP-Kzw | | DBF | 7.0 | 1524 | 165 | −155 | 919 | 709 | 15 | 32 | 187 |
| JP-Tkb | | ENF | 13.3 | 1514 | 159 | −599 | 2490 | 1781 | 45 | −7 | 579 |
| JP-Tak | | DBF | 6.8 | 2483 | 146 | −342 | 1024 | 597 | 11 | 26 | 135 |
| JP-Ta2 | | ENF | 9.8 | 1760 | 148 | −695 | 1990 | 1247 | 43 | 16 | 546 |
| JP-Tgf | | GRA | 14.3 | 1141 | 153 | −291 | 2307 | 1851 | 53 | 19 | 681 |
| JP-KaP | | CRO | 14.9 | 561 | 155 | −774 | 1127 | 584 | 70 | 7 | 894 |
| JP-Mse | | CRO | 13.9 | 1407 | 154 | −197 | 1004 | 763 | 67 | 6 | 858 |
| JP-SwL | | WAT | 11.8 | 1499 | 178 | −287 | NA | NA | 80 | 18 | 1021 |
| JP-KaL | | WAT | 16.0 | 1575 | 163 | −826 | NA | NA | 59 | 21 | 759 |
| JP-Kwg | | DBF | 15.2 | 1492 | 151 | −214 | 1631 | 1393 | NA | 18 | NA |
| JP-Shn | | MF | 12.3 | 1713 | 167 | −242 | 1612 | 1211 | 53 | 43 | 675 |
| JP-Nkm | | ENF | 0.5 | 2544 | 162 | −381 | 1442 | 725 | 36 | −3 | 445 |
| JP-Fmt | | MF | 15.0 | 1611 | 158 | 35 | 2720 | 1652 | 71 | 35 | 913 |
| JP-Kgu | urbanization | URB | 16.5 | 1400 | 149 | NA | NA | NA | 27 | 41 | 344 |
| JP-Fjy | | ENF | 9.9 | 1989 | 165 | −404 | 1772 | 1270 | 39 | 20 | 501 |
| JP-Fhk | before | DIS | 9.6 | 1846 | 168 | −74 | 1945 | 1873 | 43 | 43 | 554 |
| | thinning | DNF | 9.6 | 1846 | 168 | −433 | 1914 | 1619 | 40 | 40 | 510 |
| JP-SMF | | MF | 14.8 | 1543 | 165 | −142 | 1587 | 1059 | 51 | 23 | 658 |
| JP-Nuf | | DBF | 15.4 | 1465 | 156 | −327 | 1590 | 1139 | 22 | 16 | 277 |

| Site ID | Disturbance | Land cover | TA | PREC | Rsd | NEE | GPP | RECO | $LE$ | $H$ | ET |
|---|---|---|---|---|---|---|---|---|---|---|---|
| | | | °C | mm yr$^{-1}$ | W m$^{-2}$ | g C m$^{-2}$ yr$^{-1}$ | g C m$^{-2}$ yr$^{-1}$ | g C m$^{-2}$ yr$^{-1}$ | W m$^{-2}$ | W m$^{-2}$ | mm yr$^{-1}$ |
| JP-Tdf | | DBF | 14.8 | 2039 | 155 | −601 | 1559 | 840 | 45 | 19 | 584 |
| JP-Yms | | DBF | 15.0 | 1384 | 159 | −223 | 1644 | 1400 | 63 | 30 | 805 |
| JP-Nap | | WAT | 16.5 | 1083 | 176 | −48 | NA | NA | 60 | 9 | 773 |
| JP-Ako | | EBF | 15.3 | 739 | 169 | NA | NA | NA | 27 | 47 | 347 |
| JP-Sac | urbanization | URB | 16.4 | 1594 | 159 | 5807 | NA | NA | 28 | 43 | 354 |
| JP-Ozm | urbanization | URB | 16.5 | 1828 | 150 | 793 | 1485 | 2353 | 52 | 23 | 673 |
| JP-Om1 | urbanization | URB | 17.5 | 1202 | 165 | 1032 | 515 | 1622 | 23 | 39 | 294 |
| JP-Om2 | mowing | GRA | 16.9 | 1466 | 166 | 430 | 2634 | 2937 | 71 | 9 | 908 |
| JP-Hc3 | | CRO | 15.8 | 1136 | 175 | −663 | 1265 | 625 | 51 | 10 | 659 |
| JP-Hc2 | | CRO | 15.7 | 1141 | 161 | −132 | 890 | 697 | 60 | 10 | 772 |
| JP-Khw | | ENF | 15.2 | 2294 | 158 | −906 | 2359 | 1689 | 81 | 20 | 1044 |
| JP-Ynf | | EBF | 20.9 | 2611 | 159 | −374 | 2200 | 1808 | 65 | 5 | 834 |
| TH-Kog | | EBF | 19.9 | 2004 | 183 | −1301 | 2078 | 1178 | 73 | 23 | 935 |
| TH-Mae | | DBF | 25.3 | 1333 | 205 | −579 | 2215 | 1783 | 69 | 36 | 888 |
| KH-Kmp | | EBF | 26.9 | 1786 | 206 | −72 | 3842 | 3044 | 106 | 19 | 1368 |
| MY-LHP | | EBF | 26.2 | 2752 | 184 | −989 | 3431 | 2552 | 89 | 28 | 1157 |
| ID-Pag | | EBF | 26.1 | 2639 | 200 | −183 | 3840 | 3486 | 111 | 29 | 1430 |
| ID-PaB | fire | OSH | 26.4 | 2642 | 197 | −110 | 1746 | 1450 | 90 | 27 | 1164 |
| ID-PaD | | EBF | 26.2 | 2543 | 197 | 179 | 3100 | 2997 | 94 | 29 | 1215 |

## 3 Database summary

### 3.1 CO$_2$ flux

Based on the dataset constructed, mean seasonalities in NEE, GPP, and RECO were as expected from the biomes and mean climatology (Figs. 5, 6). In northern boreal forests in Siberia (RU-Tur, RU-NeF, RU-SkP, RU-Sk2), the magnitude of the flux was generally low, and growing seasons when GPP was not negligible were short. In the southern Eurasian boreal forests in Siberia and Mongolia (RU-Ege, MN-Udg, MN-Skt), the magnitudes of CO$_2$ fluxes were greater than those in the above northern boreal forests. Inland grasslands in Mongolia (MN-Nkh, MN-Hst, MN-Kbu) had smaller CO$_2$ flux magnitudes than the nearby forests (MN-Udg, MN-Skt). For temperate forest and grassland sites, the dataset showed known seasonality with spring onset, summer peak, and autumn senescence, with low fluxes in winter. Among forest sites, seasonal variations became smaller in the subtropics (JP-Ynf), and clear seasonality disappeared in the tropics (KH-Kmp, MY-LHP, ID, Pag, ID-PaD, ID-PaB) as the climate became warmer. Among rice paddies, single-cropping sites had a single peak (JP-Mse, JP-Hc2), but a double cropping site had two peaks (JP-Hc3) in GPP, RECO, and NEE (Fig. 6). For lakes (JP-SwL, JP-KaL), a pond (JP-Nap), and an urban center (JP-Sac), CO$_2$ fluxes showed smaller seasonality than those at vegetation surfaces.

Some data for CO$_2$ fluxes raise suspicions. First, markedly negative NEE values in harsh winters were estimated for MN-Skt and MN-Kbu (Figs. 5, 6), which could be caused by an artifact known for the open path sensor (Burba et al., 2008). The artificially negative NEE caused a considerable positive GPP in winter. Data users should be cautious about the data for MN-Skt and MN-Kbu. Second, the day-

time partitioning method extrapolated the relationship obtained during the growing season to winters when NEE was not measured. The result was erroneous estimation of GPP and RECO (e.g., JP-Nkm in Fig. 5). Using the nighttime approach, GPP and RECO were not estimated for the period when NEE was not measured. Despite these suspicious data, the fluxes partitioned using the nighttime and daytime methods were generally consistent across the sites.

The spatial variabilities in annual NEE, GPP, and RECO were also consistent with earlier reports for Asian ecosystems (Fig. 7; Table 3). In Asia, the spatial variabilities in GPP and RECO are explained mostly by the mean annual air temperature (Hirata et al., 2008; Kato and Tang, 2008; Saigusa et al., 2013; Yu et al., 2013). Except in disturbed forests and croplands, GPP and RECO increased linearly with mean annual air temperature (Fig. 7). Correlations of GPP and RECO with the annual sum of precipitation were lower than with the mean annual air temperature. No clear correlation was found between annual NEE and mean annual air temperature or annual sum of precipitation, but the maximum CO$_2$ sink (i.e., negative NEE) with each temperature range appeared to be increased by temperature up to the annual mean temperature range until approximately 10 °C (Fig. 7a). Except for disturbed forests and urban sites, most ecosystems were estimated to be a CO$_2$ sink of up to 1.0 kg C m$^{-2}$ yr$^{-1}$.

In the developed dataset, annual CO$_2$ fluxes tended to differ by land cover type (Fig. 8). Forest ecosystems included in the datasets had, on average, similar CO$_2$ sinks. Among the forest ecosystems, the mean CO$_2$ sink tended to be highest in ENF. GPP and RECO in temperate managed grasslands were higher than those in natural grasslands in Mongolia and Russia. The annual CO$_2$ sink also tended to be greater in managed grasslands compared to natural grasslands, except for

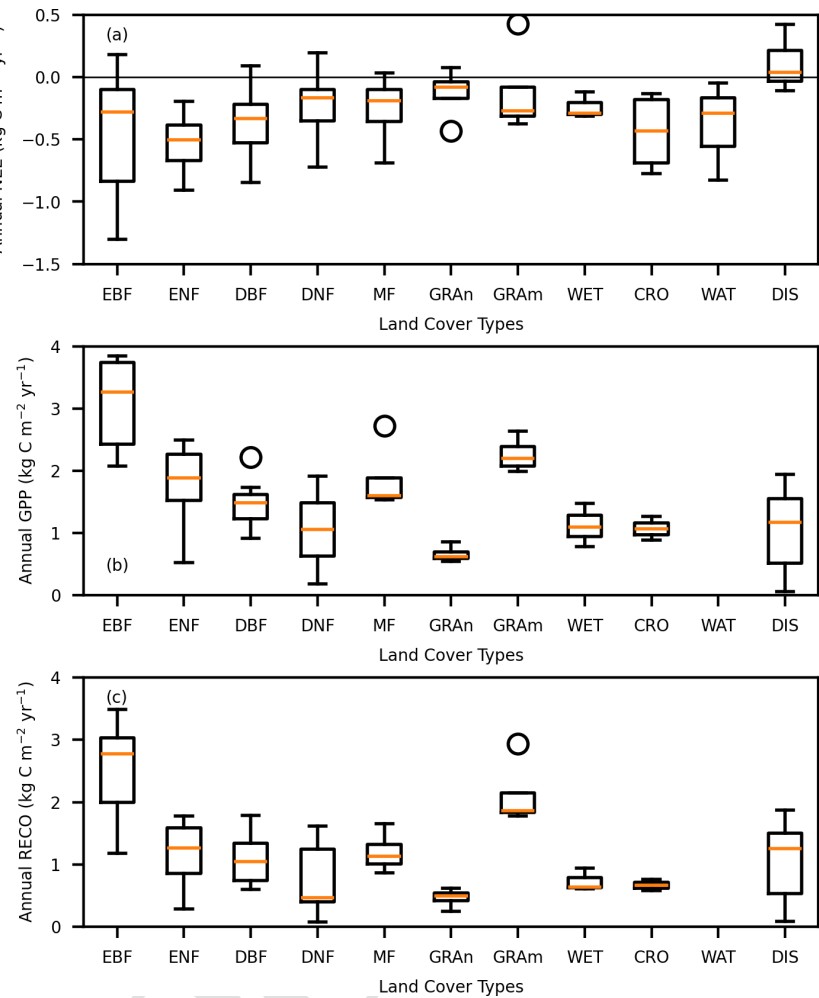

**Figure 8.** Boxplots for annual NEE, GPP, and RECO by land cover type. GPP and RECO were estimated using the daytime method. Fluxes at urban sites are not shown because the range of urban $CO_2$ emissions was totally different from those for vegetation or lakes. Because flux partitioning was not conducted for lakes and ponds, partitioned fluxes for these water surfaces were not shown. Land cover type abbreviations are in Fig. 3, although the grassland category was separated into natural grassland (GRAn) and managed grassland (GRAm). The definition of DIS was the same as in Fig. 7, where all data from RU-NeC, RU-NeB, and ID-PaB are also classified as DIS. The box represents the interquartile range (25th to 75th percentiles), the whiskers represent the maximum and minimum values, excluding outliers shown by circles, and the orange bar represents the median value.

a frequently mowed site (JP-Om2), which exhibited net annual $CO_2$ emissions. Disturbed forests, on average, acted as a small $CO_2$ source. $CO_2$ emissions in urban centers (JP-Sac; 5.8 kg C m$^{-2}$ yr$^{-1}$; not included in Fig. 8a) were considerably higher than those from natural or agricultural ecosystems. The annual GPP was highest in EBF among forest ecosystems, followed by ENF, DBF, and DNF. RECO was highest in EBF, whereas those in ENF, DBF, and DNF were similar to each other. Annual GPP and RECO varied greatly among grasslands because they included inland dry grasslands and Japan's weedy grasslands (Fig. 8b, c).

## 3.2 Energy fluxes

Mean annual energy fluxes represented in the dataset were explained better by air temperature than precipitation (Fig. 9; Table 3). The mean annual $LE$ increased with the mean annual air temperature; their strong linear correlation could be explained by a close coupling between transpiration and photosynthesis (Medlyn et al., 2011), where spatial variations in annual GPP were strongly correlated with annual air temperature (Fig. 7c). Evaporation could also be enhanced under high air temperature and resulting high VPD conditions (Zhang et al., 2016). The dataset included mostly ecosystems around the Pacific Ocean, which were especially densely distributed in Japan, whereas water-limited inland ecosystems were scarce. Consequently, the correlation between $LE$ and

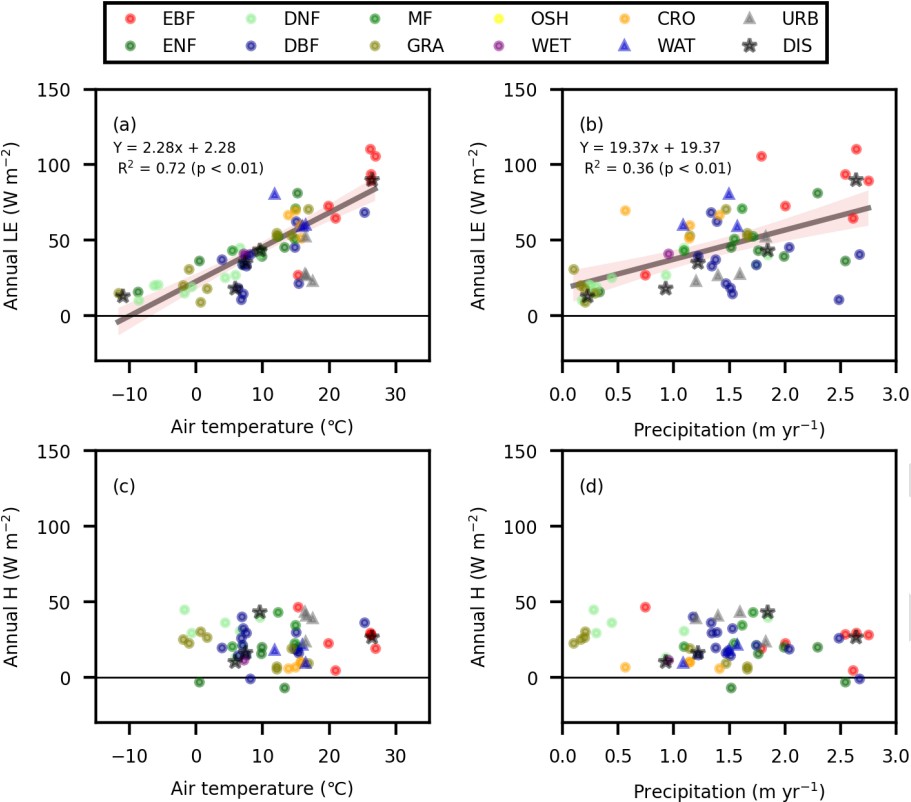

**Figure 9.** Relationships of annual energy fluxes of the latent heat flux ($LE$) **(a, b)** and sensible heat flux ($H$) **(c, d)** to the mean climate of the annual mean air temperature **(a, c)** and annual sum of the precipitation **(b, d)**. The classification of the disturbed forest (DIS) is as in Fig. 8. Annual $H$ values were calculated only for the case where there were no missing data in the mean seasonality, whereas the missing $LE$ data during the winter were considered to be 0 for boreal forests in Russia. The lines represent linear regressions, with shading showing the confidence intervals ($p < 0.05$), determined by excluding the data from DIS, urban areas (URB), and lakes and ponds (WAT). The values are shown in Table 3. Land cover type abbreviations are in Fig. 3.

precipitation was weaker than those reported in a literature survey for Asia (Kang and Cho, 2021). Under similar climate conditions, $LE$ was lower and $H$ was higher in urban landscapes compared with vegetation surfaces, in agreement with a previous report (Ueyama et al., 2021). Mean annual $H$ did not change with air temperature or precipitation, possibly caused by missing high-latitude observations owing to missing winter data (e.g., RU-Tur, RU-SkP, RU-Ege) (Fig. 5). Negative $H$ values in high-latitude ecosystems were observed owing to decreased available energy associated with the snow albedo (Nakai et al., 2013; Ueyama et al., 2020b).

## 4   Data availability

The dataset associated with this publication can be found at the ADS website (https://ads.nipr.ac.jp/japan-flux2024/TS4), where individual site data have their own DOIs. All data are available under the CC BY 4.0 copyright policy with appropriate citations of this paper. We suggest that researchers planning to use this dataset as a core dataset for their analy-

sis contact and collaborate with database developers and relevant site teams. As in the data policy of FLUXNET2015, in case of a synthesis using both CC BY 4.0 and other private data, all data should be treated as Tier Two of the FLUXNET data policy (data producers must have opportunities to collaborate and consult with data users).

## 5   Conclusions

The JapanFlux2024 dataset is the first public dataset that includes as much data as possible, both old and new, as an activity of JapanFlux. The dataset is consistent with previous synthesis studies in Asia in terms of seasonalities in $CO_2$ and energy fluxes across Japan and East Asia but substantially increased the amount of data, i.e., 83 sites with 683 site-years from 1990 to 2023. The dataset will facilitate important studies in East Asia, including Japan, such as those on land–atmosphere interactions, improvement of process models, and upscaling fluxes using machine learning. Because the dataset is processed in line with selected procedures from the FLUXNET standard dataset, the JapanFlux2024 dataset

will bridge collaborations between researchers from Asia and FLUXNET.

**Author contributions.** The JapanFlux2024 dataset was conceptualized by MU. The standardized dataset was prepared by MU and YT, and the metadata was compiled by HY and TH in collaboration with the data contributors. The data distribution website was developed by a team led by HY. KI contributed to the editing of the paper. The remaining co-authors contributed eddy covariance data to the dataset and/or participated in the editing of the paper.

**Competing interests.** The contact author has declared that none of the authors has any competing interests.

**Disclaimer.** Publisher's note: Copernicus Publications remains neutral with regard to jurisdictional claims made in the text, published maps, institutional affiliations, or any other geographical representation in this paper. While Copernicus Publications makes every effort to include appropriate place names, the final responsibility lies with the authors.

**Acknowledgements.** The development of the database was supported by the digital biosphere project under KAKENHI (21H05316, to Tomo'omi Kumagai), PAWCs project under KAKENHI (19H05668), JSPS A3 Foresight Program (JPJSA3F20220002), and Arctic Challenge for Sustainability II (ArCS II; JPMXD1420318865). The CH-Lsh data were provided by Nobuko Saigusa of the National Institute of Environmental Studies and Huimin Wang of the Chinese Academy of Sciences. Observations at JP-Tmk were supported by Ryuichi Hirata of the National Institute of Environmental Studies. Observations of JP-Kgu were supported by Manabu Kanda of the Institute of Science Tokyo. Observations of JP-Tdf were supported by Shigeaki Hattori of Nagoya University. Observations of JP-Ynf were supported by Shingo Tanigushi of the University of the Ryukyus. Masahito Ueyama was supported by KAKENHI (18H03362, 24K03065). Sachinobu Ishida was supported by KAKENHI (25450201). Kazuhito Ichii, Hiromi Yazawa, and Makiko Tanaka were supported by KAKENHI (22H05711, 22H05004, 24H01504) and the Environment Research and Technology Development Fund (JPMEERF24S12207). Hiroki Iwata was supported by KAKENHI (17H05039, 21H02315, 23K21248, 23KK0194). Michiaki Sugita was supported by KAKENHI (15K01159, 20H01384, 23K20125). Takanori Shimizu was supported by KAKENHI (20H0309). Multidisciplinary observations at Takayama sites (JP-Tak and JP-Ta2) were supported jointly by Hiroyuki Muraoka (KAKENHI 21H05316, 21H05312, 19H03301), Taku M. Saitoh (KAKENHI 18780113, 21241009, 22248017, 23710005, 24241008, 26241005, 26292092, 15H04512, 20H03041, 20K06144, 21H02245, 21H05316, 23K11395, 24K01818, 24K00986, the Environment Research and Technology Development Fund (JPMEERF20232M01) of the Environmental Restoration and Conservation Agency provided by the Ministry of the Environment of Japan, the Global Environment Research Coordination System from the Ministry of the Environment, Japan MAFF2254), Hiroaki Kondo, Shohei Murayama, Shigeyuki Ishidoya, and Takahisa Maeda (KAKENHI 24241008, 24310017, 15H02814, 18H03365, 19H01975, 22H00564, 22H05006, Global Environment Research Coordination System from the Ministry of the Environment, Japan MAFF0751, MAFF1251, MAFF2254, the Global Environment Research Fund of the Ministry of the Environment, Japan S-1: Integrated Study for Terrestrial Carbon Management of Asia in the 21st Century Based on Scientific Advancement). JP-Spp, JP-Api, JP-Fjy, JP-Yms, and JP-Khw were supported by KAKEN (16K07789), Research revolution 2002: Global Warming Initiatives (FY2002-2006) by the Ministry of Education, Culture, Sports, Science, and Technology of Japan, Commissioned project study from the Ministry of Agriculture, Forestry, and Fisheries (JPJ005317), Environment Research and Technology Development Fund (S-1), and Research Coordination System (MAFF0751, 1251, 2254) from the Ministry of the Environment of Japan, and research grants (#199903, #200303, #201802) from the Forestry and Forest Products Research Institute. JP-Tom was supported by KAKENHI (11213204, 14656059, 16208014, 2331001513, 2529207903) and by the Ministry of the Environment (0708BD437, D-0909), given to Tsutom Hiura. JP-Mse was supported by the Global Environmental Research Fund (S-1) of the Ministry of Environment of Japan, a research project entitled "Development of technologies for mitigation and adaptation to climate change in agriculture, forestry, and fisheries" by the Ministry of Agriculture, Forestry, and Fisheries of Japan, and KAKENHI (19H03077, 19H03085, 23H02341). Kazuho Matsumoto was supported by KAKENHI (25304027, 16H02762, 21H02238, 22K05752, 24H01520). Trofim Maximov was partly supported by the project "Study of biogeochemical cycles and adaptive reactions of plants of boreal and arctic ecosystems of northeastern Russia" (AAAA-A21-121012190034-2) of the Ministry of Education and Science of Russia. We thank the two anonymous reviewers and Dario Papale for their constructive comments and suggestions.

**Financial support.** This research has been supported by the Japan Society for the Promotion of Science (grant nos. 21H05316, 19H05668, and JPJSA3F20220002) and the Ministry of Education, Culture, Sports, Science, and Technology (grant no. JP-MXD1420318865). TS5

**Review statement.** This paper was edited by Hanqin Tian and reviewed by Dario Papale and two anonymous referees.

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

## Remarks from the typesetter