# Peer review of "The JapanFlux2024 dataset for eddy covariance observations covering Japan and East Asia from 1990 to 2023"

_Earth System Science Data, 2024_

## Author Comment (AC1)

**Dear Editor**

Ueyama, et al. The JapanFlux2024 dataset for eddy covariance observations covering Japan and East Asia from 1990 to 2023

We have now submitted the manuscript revised based on the comments from the three reviewers and have uploaded the updated dataset to our data portal. Each site-specific dataset is now assigned a DOI. Since the initial submission, we have added three new sites (JP-NsC, JP-NsM, and RU-USk) and included an additional year of data for JP-Tef and JP-Nuf.

In light of their substantial contributions to the updated dataset, we would like to add the following two co-authors:

Shoji Matsuura (Institute for Agro-Environmental Sciences, National Agriculture and Food Research Organization (NARO), Tsukuba 305-8604, Japan)

Takafumi Miyama (Forestry and Forest Products Research Institute, Tsukuba 305-8687, Japan)

We are confident that the revised manuscript now meets the standards of Earth System Science Data (ESSD). Once again, thank you very much for handling our manuscript. Sincerely yours, Masahito UEYAMA Osaka Metropolitan University

**Responses to reviewer's comments. (Reviewer 1)**

Ueyama, et al. The JapanFlux2024 dataset for eddy covariance observations covering Japan and East Asia from 1990 to 2023

**Revisions are colored red.**

We would like to express our sincere gratitude for the constructive and insightful comments, as well as the kind support provided to improve the dataset. In response to the reviewer's suggestions, we have reprocessed the dataset, reconstructed the metadata, updated the webpage, and added DOIs for each site. As an additional enhancement, we have added three new sites (JP-NsC, JP-NsM, and RU-USk), as well as one extra year of data for JP-Tef and JP-Nuf. As a result, the dataset now consists of 83 sites with 683 site-years. We believe that these revisions have significantly enhanced both the manuscript and the dataset, making them more valuable to the FLUXNET community. We are confident that the revised manuscript now meets the standards of ESSD. Once again, thank you for your thoughtful and valuable feedback.

First, I don't need to be anonymous: Dario Papale.

This is a paper that a lot of people were waiting for, and I think it is an incredible contribution toward the FLUXNET growth and the inclusion of JapanFlux in the international collaboration already happening. I appreciate the effort and the willingness, and the dataset is of very high interest, no doubts that it should be published in this journal. Asian contribution in FLUXNET was very limited until now and this dataset really move Japan and its very nice community upfront in the idea of an open and accessible FLUXNET. Congratulations and thanks for this.

We are pleased to contribute to FLUXNET science by providing our JapanFlux dataset. Thank you, Dario-san.

That said, my comments are more related to the formats and procedures. I have no specific comments on the way the authors decided to process the data, there are multiple options of processing and they selected one (REddyProc, specific site management and exceptions etc.). However, what can really cause misinterpretation, confusion and misuse of the data by the users is the consistency of products.

In particular, the authors decided (really nice in principle and appreciated) to follow the FLUXNET2015 standard, that is obtained with the ONEFlux pipeline (Pastorello et al. 2020), that is the same used also by AmeriFlux, ICOS, the European Database and recently OzFlux. The problem is that following a standard means that format, content and processing are the same (Interoperability). I checked carefully the products for a number of sites and the main issue I found is that the meaning of the variables (with the same name), but also the file content and the list of variables are different between this collection and the standard FLUXNET/ONEFlux. Some examples:

Thank you for highlighting these important issues. We acknowledge that the current dataset is not fully compatible with FLUXNET2015. Accordingly, we have revised the manuscript to avoid referencing compatibility throughout the text and have redefined the variable basename (as shown in the new Table 2). A detailed description of the revisions is provided below. The way USTAR filtering is managed is completely different. In the ONEFlux two methods are used and a joint population of threshold is used to estimate uncertainty, here this is not happening and one method is used and 3 percentiles extracted.

The meaning of the NEE\_REF is completely different, since in the ONEFlux is extracted among 40 realizations and based on the NEE timeseries, here is based on one threshold of USTAR

The energy balance corrected version of the H and LE is not produced in this colelction

There are variables that do not exist in the FLUXNET standard (neither in the FULLSET or SUBSET), like all the variables with the three position indexes (e.g. TA\_1\_1\_1, that are very important but not included in the standard FLUXNET files), or variables such "RH\_multiple" and "SW\_IN\_SLOPE\_PI\_1\_1\_1", or some QC that is used with a different meaning (e.g. to select fluxes based on wind sectors)

The content and split of variables between the FULLSET and the SUBSET is not the same as the FLUXNET/ONEFlux and follows a different logic

The processing of sites where either the USTAR filtering or the partitioning failed is also fundamentally different respect to what is done in the FLUXNET/ONEFLux pipeline

The inclusion of CH4 is also something not present currently in the FLUXNET/ONEFlux pipeline All this makes the statement "The JapanFlux2024 dataset is compatible with the datasets provided by

FLUXNET (e.g., FLUXNET2015)." at line 116 not true and valid.

Now, I need to be very clear on this: I'm not criticizing the approach of the authors or promoting what is done in FLUXNET/ONEFlux or suggesting to use ONEFlux. My only concern is that different things should be named in different ways to avoid confusion in the users.

For this reason, my suggestion and request would be to not use the official FLUXNET naming of files and also of variables that are not exactly produced in the same way; just use a different naming structure that you define in the paper and metadata associated to the data, the contribution would be still extremely valuable and a ONEFlux version of the data, if you want, can be still created later.

Regarding the u threshold, given the differences in methodology, we have chosen to use the "\_REF" label for the gap-filled flux variables. For variables unique to JapanFlux2024, we have prepared a new Table 2, which lists and defines these variables in detail. Concerning the two datasets (FULLSET and SUBSET), we believe it remains appropriate to provide both, and therefore we have decided to retain the current file naming convention. The sentence regarding compatibility has also been revised—for example, in Lines 115–116: "The JapanFlux2024 dataset is processed with the standard data processing methods employed by FLUXNET (e.g., FLUXNET2015), although some modifications and simplifications were made.".

There are two additional important points to consider, again to ensure a full compatibility and participation to the international effort, and obtain what the authors state in the conclusions: "JapanFlux2024 dataset will bridge collaborations between researchers from Asia and FLUXNET."

The metadata and BADM: it is an incredible effort and crucial information, these are very important data, thanks for collecting and sharing them. However, checking some of them I found that the data are not always following the controlled vocabularies and format requirements of the BADM system and this would create problems in case users want to mix data across networks. In this case I see two alternatives: either use a different metadata scheme or make the BADM compatible with the standard (see below)

The BADM files have been fully revised, and re-uploaded to the JapanFlux2024 portal site.

The FLUXNET site codes: the authors assigned FLUXNET codes to the sites, however I see two main issues here, again that can affect users: 1) some of the sites already participated in the past to data collections (LaThuile 2008, FLUXNET2015, FLUXNETCH4) and they already had a FLUXNET code.

These codes are supposed to the persistent and unique so in this case the same code should be used. Examples: JP-Tky, already shared as JP-Tak, JP-Tmk (JP-Tom), JP-Tse (JP-Tef), RU-Spl (RU-SkP) and possibly others (CN-Qhb-CN-Ha2, ID-Puf-ID-Pag...). It should also be noted that the FLUXNET codes (agreed across regional networks) are case sensitive (so for example JP-Smf was already JP-SMF and this last should be used). 2) in the general FLUXNET code standard system, the small "L" and the capital "i" letters and not any more used, for the confusion between them in some fonts (I and l). Examples: JP-Ksl, JP-Swl, MN-Udl...

The site codes that did not follow naming rules have been redefined.

On these two points, as already discussed in the past in the context of the Regional Networks collaborations between ICOS/European Database and the AsiaFlux and JapanFlux offices, I can provide assistance to check the BADM and automatically identify inconsistencies with the standard and to support the FLUXNET codes assignment and check against already used codes. All this clearly keeping myself external to the paper due my reviewer role.

Thank you for the helpful check for the site codes, BADM after this review.

**Additional points:**

Did the authors verified all the geographic coordinates? This is a very important point in particular for the remote sensing link

We have recheck the geographic coordinates.

Line 220: not clear the meaning of the sentence "Calculation of GPP and RECO by the daytime partitioning method was based on the parameterized model, which did not directly use observed NEE.". The NEE are used for the parameterization of the model, like for the nighttime method.

We have modified sentence in Lines 232-233: "GPP and RECO using the daytime partitioning method were calculated based on a fitted model that combines a light-response curve and a temperature-dependent respiration model; thus, the daytime method did not directly add up observed NEE (Wutzler et al., 2018).".

Thank you for your great contribution to FLUXNET

We, JapanFlux, are pleased to contribute our data to FLUXNET.

**Responses to reviewer's comments. (Reviewer 2)**

Ueyama, et al. The JapanFlux2024 dataset for eddy covariance observations covering Japan and East Asia from 1990 to 2023

**Revisions are colored red.**

We would like to express our sincere gratitude for the constructive and insightful comments, as well as the kind support provided to improve the dataset. In response to the reviewer's suggestions, we have reprocessed the dataset, reconstructed the metadata, updated the webpage, and added DOIs for each site. As an additional enhancement, we have added three new sites (JP-NsC, JP-NsM, and RU-USk), as well as one extra year of data for JP-Tef and JP-Nuf. As a result, the dataset now consists of 83 sites with 683 site-years. We believe that these revisions have significantly enhanced both the manuscript and the dataset, making them more valuable to the FLUXNET community. We are confident that the revised manuscript now meets the standards of ESSD. Once again, thank you for your thoughtful and valuable feedback.

The AmeriFlux, EuroFlux, and OzFlux networks have been able to ensure the accessibility and usability of high-quality, long-term ecosystem flux measurement data, which are essential for regional ecosystem studies, modeling, and multi-site synthesis. This has been made possible through the support of the AmeriFlux Management Project (AMP), Integrated Carbon Observation System (ICOS), and Terrestrial Ecosystem Research Network programs (TERN), which collaborate with the principal investigators of each flux observation site.

However, unlike other regional networks, AsiaFlux lacks a dedicated support program, resulting in its database being limited in both scale and update frequency. This presents one of the challenges in using eddy covariance flux data for a bottom-up approach to estimating the global greenhouse gas (GHG) inventory, despite the advantage of the eddy covariance method in directly measuring GHG fluxes rather than concentrations.

This data paper by Ueyama et al. is expected to be a breakthrough in overcoming the limitations of flux data sharing in the Asian region. JapanFlux2024 will serve as a role model for flux observation networks in other Asian countries, and I hope that flux datasets from various Asian countries will continue to be released in a series. Ultimately, I look forward to the rapid launch of a new AsiaFlux database that integrates these national flux databases.

Thank you for your constructive comments and encouragement.

*I sincerely appreciate the authors' efforts and provide the following minor comments for further improvement of the manuscript:*

1. L140-141: Most of the sites were established for long-term monitoring of  $CO_2$  fluxes, but intensive observations for about a week in the 1990s were also included in the dataset.

--> Short-term data can still hold significant value depending on the purpose. Data from short-term experiments can be utilized in various ways. In particular, datasets collected before the widespread adoption of the eddy covariance system may be of great importance.

Thank you for your constructive comments and encouragement.

2. L186-187: If only the rainfall was measured, the correction ratio was determined using liquid precipitation, which was defined as precipitation when air temperature was greater than  $0 \,^{\circ}$ C.

--> When distinguishing precipitation as either rain or snow, I recommend considering relative humidity (RH) in addition to air temperature (Matsuo et al., 1981).

Matsuo, T., Sasyo, Y., & Sato, Y. (1981). Relationship between types of precipitation on the ground and surface meteorological elements. Journal of the Meteorological Society of Japan. Ser. II, 59(4), 462-476.

Based on the suggestion, we have reprocessed the partitioning of rainfall and snowfall and have reconstructed the precipitation data. We have added the sentences in Lines 188-191 as "If only the rainfall was measured, the correction ratio was determined using liquid precipitation which was defined as precipitation when relative humidity was below the critical relative humidity ( $RH_{cri}$ ; %):  $RH_{cri} = 92.5 - 7.5T$ , where T is the air temperature (Matsuo and Sasyo, 1981).".

3. L194-195: In this dataset, we determined the u\* threshold each year to consider its potential shift over the years, which is termed as a Variable u\* Threshold (VUT) in FLUXNET2015 (Pastorello et al., 2020).

--> Doesn't the u\* threshold vary seasonally? Are there no influences from seasonally varied meteorological conditions or phenological changes of the canopy?

Yes, the u\* threshold is fixed for each year. We have added an explanation of the threshold in Lines 197-199: "In the moving point method, the u\* threshold was first determined for each of the four seasons, and the maximum value among them was used for the entire year. Thus, the determined u\* threshold was conservative (Papale et al., 2006).".

4. L199-200: For urban sites, the threshold was generally not used (e.g., Liu et al., 2012; Ueyama and Ando, 2016); thus, the u\* filtering was not applied for highly urbanized sites (JP-Sac and JP-Kgh).

--> Is u\* filtering generally not applied to urban sites? It would be helpful to provide a brief explanation for readers who may not be familiar with urban flux observations.

We have added an explanation in Lines 207-211: "For urban sites, the threshold was generally not used for two reasons (e.g., Liu et al., 2012; Ueyama and Ando, 2016): 1) nighttime CO2 fluxes were not expected to correlate with air temperature, making it difficult to evaluate the correct u\* threshold, and 2) the surface layer was often unstable even at night. Consequently, the u\* filtering was not applied for highly urbanized sites (JP-Sac and JP-Kgu).".

5. L208-209: H, LE, and NEE were filled with the four different u\* thresholds (reference, 5th, 50th, and 90th percentile values) using MDS.

--> Does this apply only to nighttime data? Is it typical practice to apply  $u^*$  thresholds to H and LE as well? Given that turbulent mixing is closely related to aerodynamic and radiative coupling, isn't applying  $u^*$  thresholds more complicated for H and LE compared to  $CO_2$  flux? Were the  $u^*$  thresholds used here derived from nighttime  $CO_2$  flux? More details would be helpful.

Thank you for raising these important points. First, the u\* filtering was applied only during nighttime, following the methodology provided by REddyProc, as scalar transport is effective

during the daytime under unstable conditions, even in calm conditions. The u\* threshold was determined using nighttime CO2 fluxes, which is described in Lines 196-197: "The u\* threshold was determined from the temperature sensitivity of nighttime net ecosystem exchange (NEE)". Second, the u\* filtering was applied only to CO2 flux and not to H and LE, in line with the current FLUXNET approach. In the revised dataset, we have reprocessed the H and LE data without applying the u\* filtering. To clarify the processing, we have added the following sentences in Lines 220-221: "NEE were filled with the four different u\* thresholds (reference, 5th, 50th, and 90th percentiles values) using MDS, whereas H and LE were filled without applying the u\* threshold.".

6. L305-306: A quality information flag was provided for gap-filled variables, where 0 is the original data, 1 is a gap-filled value of the most reliable quality, 2 is a gap-filled value of medium quality, and 3 is a gap-filled value of the least reliable quality.

--> It would be useful to include a brief explanation of how the quality levels for gap-filled data are determined.

We have added an explanation in Lines 322-325: "A quality information flag was assigned for gap-filled variables, where 0 is the original data, 1 is a gap-filled value of the most reliable quality (calculated using a 14-day window), 2 is a gap-filled value of the medium quality (calculated using a 14- to 56-day window), and 3 is the gap-filled value of the least reliable quality (calculated using a window longer than 56 days) (Wutzler et al., 2018).".

7. L368-369: The maximum  $CO_2$  sink (i.e., negative NEE) with each temperature range appeared to increase with temperature up to the annual mean temperature of approximately 15°C.

--> Where can this result be found?

Since this is a data paper, we did not conduct statistical analysis. The CO2 sink appears to be highest in areas where the annual air temperature is approximately  $15^{\circ}$ C (Fig. 7a). We understand that referencing  $15^{\circ}$ C may seem overinterpreted, so the sentence has been revised in Lines 387-389: " the maximum CO2 sink (i.e., negative NEE) with each temperature range appeared to be increased by temperature up to the annual mean temperature range until approximately  $10^{\circ}$ C (Fig. 7a).".

8. What does the circle in Figure 8(b) represent?

We have added an explanation in Lines 444-446: " The box represents the interquartile range (25th to 75th percentiles), the whiskers represent the maximum and minimum values, excluding outliers shown by circles, and the orange bar represents the median value.".

9. L447-449: As in the data policy of FLUXNET2015, in case of a synthesis using both CC BY 4.0 and other private data, all data should be treated as Tier Two of the FLUXNET data policy (data producers must have opportunities to collaborate and consult with data users).

--> Besides the JapanFlux2024 dataset, which is publicly available under the CC BY 4.0 copyright policy, are there other datasets uploaded on the ADS website?

All data for JapanFlux2024 is licensed under CC BY 4.0; therefore, Tair 2 data is not included in the dataset.

10. https://ads.nipr.ac.jp/japan-flux2024/

--> It would be beneficial to include IGBP classifications in the site list table, as this information is

highly relevant for data users.

We have added the IGBP classifications to the site list on the website.

**Responses to reviewer's comments. (Reviewer 3)**

Ueyama, et al. The JapanFlux2024 dataset for eddy covariance observations covering Japan and East Asia from 1990 to 2023

**Revisions are colored red.**

We would like to express our sincere gratitude for the constructive and insightful comments, as well as the kind support provided to improve the dataset. In response to the reviewer's suggestions, we have reprocessed the dataset, reconstructed the metadata, updated the webpage, and added DOIs for each site. As an additional enhancement, we have added three new sites (JP-NsC, JP-NsM, and RU-USk), as well as one extra year of data for JP-Tef and JP-Nuf. As a result, the dataset now consists of 83 sites with 683 site-years. We believe that these revisions have significantly enhanced both the manuscript and the dataset, making them more valuable to the FLUXNET community. We are confident that the revised manuscript now meets the standards of ESSD. Once again, thank you for your thoughtful and valuable feedback.

JapanFlux2024 is a highly significant contribution to FLUXNET, greatly expanding the amount of available eddy covariance data in a part of the world that has historically been under-represented. Thus, I thank and applaud the authors for their valuable contribution to the community.

Since the two other reviewers have also posted detailed comments regarding the compatibility between this product and other FLUXNET products (thank you Dario) and the data processing itself, I will focus my review more on the readability and interpretation of the manuscript. I found the manuscript very well-written, logical, and the database summary was useful. My comments are mostly related to clarification:

Thank you for your constructive comments and encouragement.

Figure 2: I find the red circles hard to see on the green background, and it would be particularly difficult for people with red/green colorblindness. I suggest using a different marker color, and potentially filling in the markers.

We have revised Fig. 2 by changing the plot color to black and increasing the transparency of the background map.

Figure 4: To improve clarity I suggest renaming the x-axis to "years of data" and the y-axis to "Number of sites".

We have changed the label names for the x-axis and y-axis of Fig. 4, as suggested.

*Line 176: How were data aggregated if there were multiple measurements of the same parameter? Were the measurements averaged?*

We have added sentences explaining the aggregation in Lines 176-179: "If meteorological variables for multiple sensors or positions were available, these variables were prioritized and aggregated; if data were missing in the highest priority dataset, they were filled with values from the second-highest priority dataset, or, if that were also unavailable, based on the priority order. The gaps in the aggregated meteorological variables were then filled with ERA5 data

because measured variables were less biased than ERA5, even when measured at different locations within a site.".

Line 181: If you calculate vapor pressure from RH, how can you fill gaps in RH with vapor pressure?

We have added sentence explaining the gap-filling of RH in Lines 183-184: "Water vapor pressure was calculated from the relative humidity, and the gaps in relative humidity were filled using the gap-filled water vapor pressure and air temperature, rather than directly filling the relative humidity.".

Line 182: "If all meteorological variables were missing in some years, the bias was corrected using a regression for the entire data record" – please explain this in more detail, it isn't clear to me what you mean.

We have modified sentence explaining the gap-filling in Lines 184-186: "If all meteorological variables were missing in some years when constructing the linear regression, the bias was corrected using a regression for the entire multi-year data record.".

Line 193: Provide a citation for your u-star threshold determination.

We have used the method proposed by Papale et al. (2006) and added sentences explaining the u\* threshold in Lines 196-203: "The u\* threshold was determined from the temperature sensitivity of nighttime net ecosystem exchange (NEE) by seasonal clustering, an approach that is widely used in the FLUXNET community. In the moving point method, the u\* threshold was first determined for each of the four seasons, and the maximum value among them was used for the entire year. Thus, the determined u\* threshold was conservative (Papale et al., 2006). In this dataset, we determined the u\* threshold each year to consider its potential shift over the years, which is termed as a Variable u\* Threshold (vUT). The vUT differs slightly from the definition of the Variable u\* Threshold (VUT) in FLUXNET2015 (Pastorello et al., 2020) (Table 2), where VUT in FLUXNET2015 was determined by pooling data from each year along with data from the immediately preceding and following years (if available)."

*Line 229: It would seem more logical to also filter nighttime fluxes by wind direction so filtering is consistent across the data product, even if that means a greater loss in data.*

For JP-Khw, we did not filter the nighttime data using the flux footprint for two reasons: 1) rejecting much nighttime data would have increased the uncertainties associated with gap-filling, and 2) there were no clear differences in nighttime fluxes among the wind sectors. The PIs of JP-Khw also agreed with this decision. However, since the dataset includes both filtered and unfiltered data, users can choose to reprocess the data in different ways. We have added the following sentence in Lines 241-243: "To extract these flux data, daytime fluxes for a wind sector on the right bank were selected, but nighttime fluxes for all wind sectors were used to increase data availability because there were no clear differences in nighttime fluxes among wind sectors ".

*Line* 285 – *this is a tiny thing, but it should be* [FIRST\_YEAR]-[LAST\_YEAR] *instead of* [FIRST\_YEAR]\_[LAST\_YEAR] (*dash instead of underscore*)

We have modified the description based on the suggestion.

Line 399 – if you artificially set GPP, RECO, and NEE to zero to produce mean annual values, how do you define the start/end of winter? I suggest putting an asterisk next to the MN-Skt and MN-Kbu values in question and include a more in-depth explanation of how you determined when to set data

**to zero.**

We have added sentence in Lines 419-421: "GPP, RECO, and NEE at MN-Skt and MN-Kbu were also considered zero during winter, when daily mean air temperature was below -5 °C, to mitigate the influence of the negative values of  $CO_2$  fluxes caused by an artifact associated with an open-path sensor.". We have also added an asterisk next to the values of NEE, GPP, and RECO for MN-Skt and MN-Kbu.

*Line 407-408 If the continually-mowed site significantly changes your box plots, you could consider separating that site out into it's own category.*

We have separated the grassland classification into the natural and managed grassland classification in the boxplot in Fig. 8. The sentence has been revised in Lines 429-431: "GPP and RECO in temperate managed grasslands were higher than those in natural grasslands in Mongolia and Russia. The annual  $CO_2$  sink also tended to be greater in managed grasslands compared to natural grasslands, except for a frequently mowed site (JP-Om2), which exhibited net annual  $CO_2$  emissions.".

*Line* 424 – *evaporation is also coupled with air temperature, not just transpiration.*

We have added sentence in Lines 452-453: "Evaporation could also be enhanced under high air temperature and resulting high VPD conditions (Zhang et al., 2015).".

Figure 8. Please clarify what the box plots and whiskers represent (are whiskers set at 1.5 times the interquartile range?)

We have added an explanation in Lines 444-446: "The box represents the interquartile range (25th to 75th percentiles), the whiskers represent the maximum and minimum values, excluding outliers shown by circles, and the orange bar represents the median value.".

Table 2 – why are many of the values N/A? Please explain.

We have added an explanation in Lines 421-423: "The N/A values were listed because missing observations, even after gap-filled fluxes, prevented the calculation of annual fluxes or because the standard flux partitioning was not available for pond, lakes, and urban landscapes.".

---

## Author Response (AR2)

**Responses to reviewer's comments. (Reviewer 1)**

Ueyama, et al.
The JapanFlux2024 dataset for eddy covariance observations covering Japan and East Asia from 1990 to 2023

**Revisions are colored red.**

*First let me thank the authors for the consideration of all the requests and suggestions. I clearly still think that this is a great dataset and step forward. I checked some of the new dataset and revised the description and I think there are still few aspects to improve before publishing the paper, in particular related to the confusion that this dataset can create respect to the standard FLUXNET collections.*

*I acknowledge that the authors have already done a lot in this direction (e.g. o the site codes, now perfectly aligned with the FLUXNET standards so that they can be used also in other contexts) but these are the points that I would like to have better defined and stressed in the manuscript and in the data:*

> Thank you for your prompt review and valuable suggestions for improving the dataset. We have revised the dataset, including the file naming, and clarified the description of how the current dataset differs from the FLUXNET product. We believe that the revised dataset and manuscript meet the standards of ESSD and provide valuable resources for users, while ensuring no confusion with FLUXNET products.

*1) The data: the files are in a standard format, clear and well explained. However I still see a major source of confusion the use of a filename structure that is imitating the standard FLUXNET produced by ICOS, AmeriFlux and OzFlux (using the same code) but not following the same standard in the content. I agree that having a file will less variables can be useful but the name should avoid confusion for the users. You could use something like "ALLVARS" and "SELECTEDVARS", but please avoid FULLSET and SUBSET. This would be even more critical if at one point JapanFlux will decide to share data in the standard FLUXNET, where the files would have then the same exact name (but different content).*

> We apologize for the incomplete revision regarding this issue in the previous version. Following the reviewer's suggestion, we have revised the file naming convention as follows: 1) changing "FULLSET" to "ALLVARS," and 2) changing "SUBSET" to "COREVARS."

*2) The sentence on line 66 "The data is processed with the standard data processing methods employed by the FLUXNET community", repeated also in line 108 should mention that a "selection of the standard processing methods" have been used. At the same time, the same sentence at line 115-116 should be removed or changed, because the processing applied is fundamentally different (in a number of aspects, starting from the meaning and selection of the REF, the data QAQC, use of the bootstrapping for the uncertainty estimation, the EBC., how the ustar threshold were applied etc.) respect to the FLUXNET2015. In other words you should help the user to not get confused… In the current version, how a user would understand that the REF NEE in this dataset is completely different from the REF NEE in FLUXNET?*

> Following the reviewer's suggestion, we have revised the sentences in Lines 65-67: "The data was processed using selected standard methods from the FLUXNET community, with adaptations specific to the JapanFlux2024 dataset.", and in Lines 114-115: "The JapanFlux2024 dataset is processed using selected standard methods from the FLUXNET

community, with adaptations specific to the JapanFlux2024 dataset.” In conclusion the sentence have also been revised in Line 488: as ” the dataset is processed in line with reference to selected procedures from the FLUXNET standard dataset”.

*3) Still on the same point, the sentence “although some modifications and simplifications were made” is where instead more details should be provided, starting from the fact that more freedom in the data cleaning was agreed for the PIs, to the difference in the uncertainty estimation, ustar filtering and energy balance closure evaluation. You should also mention clearly that while FUXNET2015 and what is now produced by ICOS and AmeriFlux is based on the ONEFlux code, this dataset is based on a different code.*

Following the reviewer's suggestion, we have added a sentence in Lines 140-143: “The JapanFlux2024 dataset differs from datasets such as FLUXNET2015 in that it provided site principal investigators (PIs) with increased flexibility in data screening. When clear anomalies were identified, quality control procedures were applied by the management team in collaboration with the respective site PI.”.

The different approach for $u^*$ filtering was described, but following sentence has been added in Lines 203-213: “The $u^*$ threshold was determined with 100 bootstrap replicates, where reference (original data obtained without using a bootstrapped sample), the 5th, 50th, and 95th percentiles of the estimated $u^*$ threshold were used for subsequent data filtering, gap-filling, and flux partitioning. Here, the nighttime was defined as downward shortwave radiation < than 10 W m$^{-2}$, and was further confirmed using exact solar time at the site location. On the basis of the estimated $u^*$ threshold, nighttime $CO_2$ fluxes and/or NEE were eliminated. This dataset does not include the estimation of NEE using the constant $u^*$ threshold (CUT), nor the advanced uncertainty estimation provided with the _REF suffix, as implemented in the ONEFLUX pipeline (Pastorello et al., 2020). For urban sites, the threshold was generally not used for two reasons (e.g., Liu et al., 2012; Ueyama and Ando, 2016): 1) nighttime CO2 fluxes were not expected to correlate with air temperature, making it difficult to evaluate the correct $u^*$ threshold, and 2) the surface layer was often unstable even at night. Consequently, the $u^*$ filtering was not applied for highly urbanized sites (JP-Sac and JP-Kgu).”.

The difference approach for the energy balance correction was described in Lines 225-227: “In the data collected from the site teams, energy imbalance correction (Twine et al., 2000) was not applied for H and LE at any sites; thus, the gap-filled H and LE were not corrected for the energy balance closure.”.

*Basically, the fact that this important dataset is NOT compatible with the FLUXNET is not a problem in my opinion. The authors defined a product and its value is independent of the compatibility with other products. Presenting it as compatible while it is not is instead a problem that must be carefully avoided, because would impact users and future products.*

*I already signed the first review so the authors can contact me in case something is not clear of if they want to discuss specific points.*
*Dario Papale*

Thank you for your valuable feedback on the dataset and manuscript. In addition to the revisions mentioned above, we have also updated the coordinates for the sites CN-In5, CN-In6, CN-In8, and CN-In7. We believe that the ambiguities in the dataset and manuscript, which could lead to user confusion, especially regarding the differences with FLUXNET products, have been resolved, and that this product will now contribute to the community.